# *Drosophila melanogaster* Mitochondrial Carriers: Similarities and Differences with the Human Carriers

**DOI:** 10.3390/ijms21176052

**Published:** 2020-08-22

**Authors:** Rosita Curcio, Paola Lunetti, Vincenzo Zara, Alessandra Ferramosca, Federica Marra, Giuseppe Fiermonte, Anna Rita Cappello, Francesco De Leonardis, Loredana Capobianco, Vincenza Dolce

**Affiliations:** 1Department of Pharmacy, Health, and Nutritional Sciences, University of Calabria, 87036 Arcavacata di Rende (Cosenza), Italy; rosita.curcio@unical.it (R.C.); federica.marra@unical.it (F.M.); annarita.cappello@unical.it (A.R.C.); 2Department of Biological and Environmental Sciences and Technologies, University of Salento, 73100 Lecce, Italy; paola.lunetti@unisalento.it (P.L.); alessandra.ferramosca@unisalento.it (A.F.); 3Department of Biosciences, Biotechnologies and Biopharmaceutics, University of Bari, 70125 Bari, Italy; giuseppe.fiermonte@uniba.it (G.F.); francesco.deleonardis@uniba.it (F.D.L.)

**Keywords:** membrane transport, mitochondrial carrier, mitochondrial transporter, mitochondrial metabolism, *Drosophila melanogaster*, mitochondria, SLC25, proteoliposomes

## Abstract

Mitochondrial carriers are a family of structurally related proteins responsible for the exchange of metabolites, cofactors and nucleotides between the cytoplasm and mitochondrial matrix. The in silico analysis of the *Drosophila melanogaster* genome has highlighted the presence of 48 genes encoding putative mitochondrial carriers, but only 20 have been functionally characterized. Despite most Drosophila mitochondrial carrier genes having human homologs and sharing with them 50% or higher sequence identity, *D. melanogaster* genes display peculiar differences from their human counterparts: (1) in the fruit fly, many genes encode more transcript isoforms or are duplicated, resulting in the presence of numerous subfamilies in the genome; (2) the expression of the energy-producing genes in *D. melanogaster* is coordinated from a motif known as Nuclear Respiratory Gene (NRG), a palindromic 8-bp sequence; (3) fruit-fly duplicated genes encoding mitochondrial carriers show a testis-biased expression pattern, probably in order to keep a duplicate copy in the genome. Here, we review the main features, biological activities and role in the metabolism of the *D. melanogaster* mitochondrial carriers characterized to date, highlighting similarities and differences with their human counterparts. Such knowledge is very important for obtaining an integrated view of mitochondrial function in *D. melanogaster* metabolism.

## 1. Introduction

*Drosophila melanogaster*, commonly known as the fruit fly, is a useful model organism for studies on aging. It is relatively easy to culture, has a conveniently short lifespan, consists of mostly post-mitotic tissues (except for the reproductive system and a small part of the gut), and has a fully sequenced genome and well-characterized genetics. This insect uses many mitochondrial transporters homologous to the human carriers, as well as the same mechanisms for transporting several metabolites from cytosol to mitochondria and vice versa. Furthermore, it offers a variety of powerful molecular genetics methods for studying transporters, many of which would be difficult to test in mammalian models.

Mitochondrial carriers (MCs) are intrinsic proteins of the inner mitochondrial membrane, whose function is to connect the cytosol and mitochondrial matrix, by catalyzing the transport of a large number of metabolites essential for cell functions [1]. MCs represent an important link between metabolic reactions occurring inside mitochondria and those taking place outside them. Mitochondrial carriers are essential in many biochemical processes, such as the citric acid cycle, oxidative phosphorylation, the transfer of NADH and NADPH reducing equivalents, gluconeogenesis, and amino acid and fatty acid metabolism, as well as in mitochondrial duplication, transcription and translation processes, calcium-mediated cell signaling pathways and insulin secretion [1,2,3].

The existence of these carriers was demonstrated in the 1970s by transport studies carried out firstly on intact mitochondria [4] and then using artificial membranes (liposomes), in which individual isolated and purified proteins were functionally reconstituted [5].

The first carrier whose primary structure has been determined is the adenine nucleotide carrier; subsequently, the amino acid sequences of the uncoupling protein, phosphate carrier, α-ketoglutarate/malate carrier tricarboxylic acid carrier, α-ketoglutarate/malate carrier, tricarboxylic acid carrier and carnitine/acylcarnitine carrier have been determined using proteins purified from mitochondria [2,3,6,7,8]. Some carriers (e.g., those carrying adenine nucleotides, phosphate, pyruvate and glutamate) are present in all mitochondria. Others are tissue-specific and have a limited distribution, reflecting their importance in particular metabolic functions, such as urea or fatty acid synthesis [2]. Few members are instead located in other cell organelles including chloroplasts, peroxisomes and mitosomes [9,10]. Structural and functional analyses of mitochondrial carriers have highlighted some common characteristics preserved in different organisms; indeed, the polypeptide chain of each carrier has a length of about 300 amino acids, and it is organized in a tripartite structure consisting of three homologous domains, each composed of about 100 amino acids [6,11]. Each domain is formed by two transmembrane segments (each having an α-helical structure), linked by a hydrophilic loop protruding into the mitochondrial matrix and characterized by the presence of a typical signature motif (PX[DE]XX[RK] (PROSITE PS50920 and PFAM PF00153)) [6,11]. These domains are linked by short hydrophilic sequences exposed to the intermembrane space. Furthermore, the amino- and carboxy-terminal regions of the polypeptide chain protrude into the intermembrane space, as demonstrated by transmembrane topography studies [12]. In humans, 53 hypothetical genes have been identified encoding mitochondrial transport proteins to date; they are grouped in a single family called solute carrier family 25 (SLC25) [2]. The SLC25 family belongs to the largest human solute-carrier (SLC) superfamily, organized into 65 gene families and including over 400 membrane proteins catalyzing the transport of metabolites across biological membranes [13,14]. Most of these carriers have been structurally and functionally characterized. By screening *D. melanogaster* genome, 48 genes have been detected encoding putative mitochondrial carriers (Appendix A), 20 of which have been functionally characterized to date. In Table 1, a list of these genes is indicated, showing, for each one, the corresponding human gene, while a phylogenetic tree highlighting their gene products is reported in Figure 1. *D. melanogaster* proteins have sequence homology ranging from 50% to 70%, when compared to the human carriers, with the exception of uncoupling proteins, which have sequence homology of about 27–40%.

The structures of *Drosophila* and mammalian genes are different [16,17], since in this insect, many forms of alternative splicing have been detected, which are absent in humans [18,19,20]. In fact, more than half of *D. melanogaster*’s spliced genes encode at least two transcripts [21]. It has been found that about 50 genes encode more than 1000 transcript isoforms each [22]. In *D. melanogaster*, this alternative splicing mechanism is very important, and it is often employed to generate proteomic diversity and to increase cellular functions [18,20,23]. A phylogenetic analysis of *D. melanogaster* mitochondrial carriers performed considering the known mammalian carriers has revealed the presence of numerous subfamilies in the fruit fly genome [24,25,26,27]. Furthermore, a first in silico analysis of *D. melanogaster*’s nuclear energy gene sequences has suggested that a single regulatory element, resulting in a genetic regulatory circuit, can coordinate the expression of the whole set of energy-producing genes in this insect [28,29]. This motif is known as Nuclear Respiratory Gene (NRG) element [28,30]. It is a palindromic 8-bp sequence (TTAYRTAA) shared by all the nuclear insect oxidative phosphorylation (OXPHOS) genes, as well as by many other nuclear genes involved in mitochondrial biogenesis and function [28,30]. Remarkably, all the Krebs cycle and OXPHOS genes supporting the respiratory chain contain one or more NRG sites. These sites are also present in genes involved in pyruvate metabolism, as well as in many genes encoding carriers and enzymes implicated in mitochondrial energy pathways [28].

At variance with the human genes encoding mitochondrial carriers, in *D. melanogaster*, OXPHOS duplicated genes are expressed to a much lower extent with respect to their parent genes, as deduced by the abundance in the public databases of ESTs derived from their transcripts; moreover, in this insect, they display a strongly testis-biased expression pattern [31,32]. It is probable that acquiring a new expression pattern would be necessary to maintain in the genome a duplicate copy of some genes [32]. In this review, we have highlighted the current understanding of the structure, biochemical features, expression pattern, subcellular localization and function of the known mitochondrial carrier family members in *D. melanogaster*, highlighting similarities and differences with the human counterparts. In addition, intra- and inter-genome comparisons have allowed a first assessment of the evolution of this protein family.

## 2. *Drosophila melanogaster* vs. Human Mitochondrial Carriers

### 2.1. The Citrate Carrier: CG6782 vs. SLC25A1

The mitochondrial citrate carrier (CIC), also named tricarboxylate carrier, in humans is encoded by the *SLC25A1* gene. The gene is spread over 2.8 kb of human DNA, and its open reading frame encodes the mature protein consisting of 298 amino acids, preceded by a presequence of 13 amino acids [33]. Studies conducted on human prostatic epithelial cells (PNT2-C2) showed the presence of a plasma membrane citrate transporter, a variant of SLC25A1 called pmCiC, an isoform expressed at the plasma membrane level [34]. This isoform is encoded by the same gene *SLC25A1* but derives from the alternative splicing affecting the first exon. The result of this alternative transcription produces the isoform pmCiC, with a total length of 318 amino acids, compared to the mitochondrial carrier CiC, which consists of 311 total amino acids. The two isoforms differ exclusively at the level of the amino-terminal region [34]. The first studies on CIC were performed in intact mitochondria, then the protein was purified from several sources, functionally reconstituted into proteoliposomes and kinetically characterized [3,35]. This carrier catalyzes an obligatory electroneutral exchange of a tricarboxylate—such as citrate, isocitrate and cis-aconitate—for another tricarboxylate, a dicarboxylate (i.e., succinate or l-malate) or PEP [2]. Its transport activity is strongly and specifically inhibited by 1,2,3-benzenetricarboxylic acid (1,2,3-BTA), which is a substrate analog, even if other inhibitors are effective [3,35]. CIC shows conserved substrate specificity and inhibitor sensitivity both in intact mitochondria and in a reconstituted system containing the purified protein [2,36]. Further studies performed using the recombinant CIC of humans, rats and silver eels reconstituted in fused membrane vesicles of Lactococcus lactis [36] or liposomes [37,38,39,40] have revealed similar substrate specificity, since these recombinant proteins are able to transport citrate, threo-isocitrate, cis-aconitate, l-malate, PEP and also succinate [36,37,38,39,40]. Physiologically, CIC exports citrate arising from sugar sources from the mitochondrial matrix to the cytosol, where ATP-citrate lyase converts it into oxaloacetate and acetyl-CoA; the latter is then used for sterol and fatty acid biosynthesis, while oxaloacetate is reduced to malate, which is converted to pyruvate by malic enzyme, thus producing NADPH [41].

In this light, CIC activity is essential for connecting carbohydrate catabolism and lipogenesis in the liver [35,41,42]. CIC is involved in many important biochemical pathways, including gluconeogenesis, insulin secretion, inflammation [43,44,45,46], protein acetylation [45,47,48] and innate immune cell function [45]. Additionally, its activity is decreased during chronic cholestasis [49], and its inhibition might be useful in nonalcoholic fatty liver disease [50], as well as in cancer resistance [51,52,53] and in apoptosis induction in cancer cells [54]. The expression of the human CIC (hCIC) is very high in the liver, but this carrier is also expressed in the pancreas and kidney [3].

By screening the *D. melanogaster* genome for the rat CIC sequence (P32089), a single gene (*CG6782*) with 65% and 62% sequence identity with the human and rat genes, respectively, was detected; it encoded a carrier for tricarboxylate ions designated *Dm*CIC. *D. melanogaster* and other known superior eukaryotic CICs have a peculiar characteristic also shared by another member of the SLC25—i.e., the phosphate carrier [39]—that is, the possession of a presequence not useful for inserting proteins into the mitochondrial inner membrane but necessary for augmenting the population of soluble precursor CICs available for import [39,55]. Mature recombinant CIC has been overexpressed in *E. coli*, purified by Ni^+^-NTA-agarose affinity chromatography, reconstituted into liposomes and kinetically characterized [56]. Additionally, *Dm*CIC works as an obligatory antiporter, exhibiting the same substrate specificity and inhibitor sensitivity already observed for rat and silver eel mitochondrial CICs [3]. In homo-exchange experiments, the *Dm*CIC half-saturation constant (Km) value for citrate is about 132 µM, while its maximal velocity (Vmax) value is about 11.75 mmoL/min/mg protein. The *Dm*CIC Km value is higher than the values measured for human [36] and silver eel CICs [3], whereas it is similar to that determined for rat liver CIC [3]. Furthermore, expression analysis performed by semi-quantitative RT-PCR has highlighted that *Dm*CIC is equally expressed in embryos, larvae, pupae and adults, indicating that its essential metabolic function is required during each developmental stage. A conserved and unexpected role for *Dm*CIC in maintaining genome stability was accidentally discovered in 2009 [47], when *D. melanogaster* mutant brain cells, either homozygous or hemizygous for the P-element-induced late lethal mutation l(3)EP3364, exhibited recurrent chromosome breakages with splinters of chromatin. The insertion of a P-element upstream from the ATG codon of the *CG6782* gene was found. This gene was called *sea* after its chromosome splinter (“scheggia” in Italian) phenotype, while its product (corresponding to *Dm*CIC) was named Sea [47]. In *sea* mutant neuroblasts, an interphase arrest leading to cell cycle delay was observed, depending on a signaling pathway involving at least the fly homolog of ataxia telangiectasia mutated (ATM)- and Rad3-related (ATR) protein kinases, as well as the ATR-interacting protein (ATRIP) [57]. Sea/*Dm*CIC expression levels proved to be decreased in mitochondrial extracts of *sea* mutants; the latter also exhibited a strongly decreased citrate/citrate exchange rate, as well as decreased cytosolic levels of citrate with respect to the wild-type ones. Accordingly, when *sea* mutant larvae were supplemented with citrate, a significant decrease in the frequency of chromosome breakages was observed, suggesting that normal cell citrate levels are essential for chromosome integrity [47]. Sea/*Dm*CIC dysfunction directly impaired histone acetylation, since histone acetyl transferase (HAT) activity was similar in both wild-type and *sea* mutant extracts. Hence, Sea/*Dm*CIC appears to play a critical role in supplying the acetyl CoA required for nucleosome histone acetylation. On the other hand, the inhibition of histone deacetylases (HDACs) could rescue the *sea* mutant chromosome break phenotype, thus confirming the connection existing among citrate deficiency, histone acetylation failure and chromosome breakage [47]. Remarkably, the inhibition of *SLC25A1* obtained by the transfection of primary human fibroblasts with siRNA duplexes against *SLC25A1* caused the same chromosome aberrations seen in Drosophila, along with a reduction of the histone H4 acetylation level.

*Dm*CIC represents the fruit fly homolog of mitochondrial hCIC according to its high sequence identity with the human and rat CICs, transport features and kinetic parameters. Furthermore, studies conducted in *D. melanogaster* have highlighted that mitochondrial CIC has an important conserved function from Drosophila to humans in maintaining chromosome stability [47].

### 2.2. The ADP/ATP Carrier: CG1683 and CG16944 vs. SLC25A4, SLC25A5, SLC25A6 and SLC25A31

The ADP/ATP carrier (AAC), also known as the adenine nucleotide translocator (ANT), is one of the most studied mitochondrial carrier family members for its essential role in cellular energy metabolism but also because it is the most abundant protein in the mitochondrial inner membrane [2]. The amino acid sequence of AAC from beef heart was the first to be determined among mitochondrial carriers. Subsequently, AAC was identified and biochemically characterized in several different species, from mammals to plants and fungi. Homologous proteins from various species share a considerably high degree of structural and sequence homology, and perform identical or similar functions [58]. The physiological role of AAC is to transport mitochondrial ATP synthesized from oxidative phosphorylation into the cytoplasm, where it acts as the main energy currency of the cell to sustain intracellular biochemical reactions necessary for maintaining cellular functions and metabolic homeostasis. After the consequent hydrolysis of ATP into ADP, this latter is transported back into the mitochondrial matrix, where it can be rephosphorylated to ATP [58]. Several studies on the functional characterization of AAC performed both on rat liver mitochondria and on recombinant human proteins established that AAC transports the free forms of ADP and ATP, with high efficiency and selectively, in a 1:1 ratio. It can also transport some small anionic solutes at much lower rates, such as phosphoenolpyruvate (PEP) and pyrophosphate (PPi), which have three or four negative charges like ADP and ATP, probably fundamental for the interactions with AAC [2,59]. Recent advances have highlighted structural changes in the transport cycle of AAC thanks to the use of specific inhibitors known as carboxyatractyloside (CATR) and bongkrekic acid (BKA), which trap the carrier in two defined conformations [6,7]. The carrier cycles between two states; in the cytoplasmic state (c-state), AAC is bound by CATR, with its binding site oriented towards the intermembrane space, whereas in the matrix state (m-state), BKA blocks the carrier, with its binding site towards the mitochondrial matrix [6,7].When ADP binds to AAC in the c-state, it induces a conformational change that switches the carrier to the m-state, leading to ADP transport across the membrane. On the contrary, in the m-state, ATP binding from the matrix side induces eversion and results in the release of ATP into the intermembrane space and its subsequent diffusion in the cytoplasm; concomitantly, the carrier comes back to its original conformation. The availability of these inhibitors has allowed great progress in the knowledge of this carrier, from which structural information on all the SLC25 family members has been deduced. The reversibility of ADP/ATP exchange is another important characteristic of AAC, depending on the concentration of the substrates in the two major compartments and on the mitochondrial membrane potential [60]. Indeed, since one cytosolic ADP (carrying three negative charges) is exchanged for one matrix ATP (with four negative charges), the exchange reaction is electrogenic and driven by the electrical component of the proton gradient. Hence, the entry of ADP into and exit of ATP from mitochondria are favored. Apart from the exchange of ADP and ATP across the inner mitochondrial membrane, AAC also exhibits an intrinsic uncoupling activity through which it acts as a regulator of mitochondrial energy production by concomitantly modulating H^+^ flows and ADP/ATP exchange [61]. AAC seems to mediate proton leak across the inner mitochondrial membrane, dissipating the electrochemical proton gradient that would otherwise drive ATP production. It occurs in tissues not expressing uncoupling protein 1 through a mechanism similar to that proven for some uncoupling proteins, which includes carrier activation by free long-chain fatty acids [61]. Furthermore, AAC is an important regulatory and a possible structural component of the mitochondrial permeability transition pore (MPTP), a channel implicated in many human diseases. Karch et al. proposed a multi-pore model in which AAC is at least one of the molecular components of MPTP, playing a key role in the maintenance of mitochondrial functionality, as well as in the regulation of the mitochondrial apoptosis pathway [62].

In humans, four different isoforms (AAC1, AAC2, AAC3 and AAC4) have been identified, encoded by four different genes (*SLC25A4*, *SLC25A5*, *SLC25A6* and *SLC25A31*, respectively), which differ in their cell type or tissue distribution, as well as in their expression patterns in different development stages and cell differentiation states. In particular, AAC1 is specific to muscle and brain tissues, AAC2 is mainly expressed in proliferative cells and AAC3 is ubiquitous, whereas AAC4 is found almost exclusively and at low levels in the testis [63]. More than one human AAC gene has been shown to be expressed simultaneously, and no particular functional differences between isoforms have been revealed. It seems that the differential expression of one isoform rather than the others depends on the metabolic state and needs of the cell [64]. Recent evidence has revealed that the expression of human AAC isoforms is closely related to the energetic metabolic properties of cells, playing a role in the switch from the oxidative to the glycolytic metabolism of cancer cells [65,66]. In this regard, aberrant expression of AAC2 is frequently observed in various cancer types; in particular, its expression was found to be related to mitochondrial uncoupling and dysfunction. When mitochondria are markedly compromised, cells are unable to sustain oxidative metabolism, and AAC2 expression seems to be necessary to switch to glycolysis by importing ATP into mitochondria and ADP into the cytosol, where it is rephosphorylated by glycolysis. The reverse activity of human AAC2 and, consequently, that of ATP synthase, which hydrolyzes ATP, contribute to the maintenance of the mitochondrial membrane potential, providing an advantage in terms of cell survival and proliferation [66,67]. Moreover, unlike the AAC1 and AAC3 isoforms, AAC2 was found to exert not only an anti-apoptotic activity but also a cytoprotective effect in cancer cells, thus contributing to cancer development [64]. Despite the different tissue distribution and expression patterns of human AAC isoforms, the kinetics of their ADP/ATP exchange activity are largely similar between homologs. The kinetic parameters have been shown to be burdened with variability, mainly because of their dependence on the energy state of the mitochondria, influencing the phosphorylation state of the endogenous adenine nucleotide pool and the membrane potential [68]. The studies conducted on humans, have revealed that the Km of AAC3 for ADP (8.4 µM) is slightly higher than those of AAC1 and AAC2 (about 3 µM), and the Vmax of AAC3 is about two-fold higher than those of AAC1 and AAC2. On the contrary, the Km value of the more recently identified AAC4 for ADP (72 μM) is much higher than that of AAC1-3, exhibiting a much lower affinity for its substrate [63].

In *Drosophila melanogaster*, AAC is encoded by two tandem duplicated genes, stress sensitive B—known as *sesB* (*CG16944*)—and *ANT2* (*CG1683*), which are transcribed from a common promoter, and their mRNAs are generated by alternative splicing. *SesB* encodes a functional AAC isoform in Drosophila adults, and it is ubiquitously expressed at high levels, whereas *ANT2* is found only at low levels, except in the testis, but its physiological role is not yet fully understood [69]. Drosophila *sesB* encodes two transcripts differing in their 3′UTR, which could determine a different intracellular distribution of AAC. Both transcripts share a mitochondrial localization motif, but it seems to be inaccessible in one transcript, determining an extra-mitochondrial localization of AAC [70]. According to this evidence, Drosophila AAC has also been detected in myofibrils of indirect flight muscles (IFM), suggesting the existence of a direct exchange of nucleotides between myofibrils and the mitochondria around them [70,71]. This adaptive mechanism could provide, directly and more efficiently, sufficient ATP for sustaining muscle contractions. Indeed, Drosophila wing muscles lack mitochondrial arginine kinase, an enzyme that transfers high energy phosphates to arginine, forming arginine phosphate (AP) in invertebrate muscles. This phosphagen system acts by transferring energy from the sites of production (mitochondria) to those of consumption (myofibrils) and operates in a way similar to that of the creatine kinase (CK)/creatine phosphate (CP) system in vertebrates [72,73]. In the latter case, AAC forms a functional unit with mitochondrial CK, participating in the cytosolic export of phosphocreatine utilized to produce energy for muscle contraction. In Drosophila IFM, the apposition of mitochondria and myofibrils and the direct involvement of AAC compensate for the lack of a cytosolic buffering phosphagen system and a mitochondrial AK [70]. *SesB* was identified in a stress-sensitive mutant that showed paralysis in response to a mechanical stress [74]. The phenotype of *sesB* mutant flies was characterized by developmental retardation, epileptic seizures, female sterility and a greatly decreased lifespan, highlighting the crucial role of this protein in cellular energy metabolism during development. Moreover, *sesB* was found to be differentially regulated in response to a restrictive temperature. Indeed, it belongs to a cluster of Drosophila genes, named temperature-sensitive genes, whose naturally induced mutation can cause the alteration or loss of protein function at a non-permissive temperature (29 °C, vs. a permissive temperature of 20 °C), leading to a large increase in fruit fly mortality. Most proteins encoded by temperature-sensitive genes, such as sesB, were found to be implicated in oxidative phosphorylation and associated with mitochondrial activities, suggesting that increased mortality is closely related to mitochondrial dysfunction and energy metabolism disruption [75]. Hence, ADP/ATP translocases are directly involved in mitochondrial functions in *D. melanogaster*, determining cellular activity and survival. It has been reported that the overexpression of sesB and ANT2 induced apoptosis in Drosophila cultured cells, suggesting that these proteins could play an important role during programmed cell death [76]. This finding is in agreement with what was found in humans, in which AAC involvement in the regulation of the mitochondrial apoptosis pathway was proven. To date, it is known that the human AAC isoforms differently influence apoptosis, AAC1 and AAC3 acting as pro-apoptotic factors and AAC2 and AAC4 as anti-apoptotic factors. However, the debate on their mechanism of action is still open. AAC could influence cell death through a cooperation with pro-apoptotic proteins, such as Bax, or through its participation in the MPTP complex [64,77]. Alteration of AAC activity has been reported in Drosophila *nebula* mutants [78,79] *Nebula* is the Drosophila homolog of human *Down syndrome critical region gene 1* (DSCR1), one of the genes involved in the establishment of the phenotype associated with Down syndrome. DSCR1 overexpression in Down syndrome fetal brain tissue seems to contribute to mental retardation in Down syndrome patients. Similarly, Drosophila mutants overexpressing *nebula* exhibited severe learning defects and long-term memory deficiency, because of the disruption of mitochondrial function and integrity. It has been demonstrated that *nebula* is located in mitochondria, where it interacts with AAC, influencing its activity. Furthermore, several mitochondrial defects in *nebula* mutants were found to be similar to those observed in *sesB* mutants. This evidence suggests that *nebula* and *sesB* could play a role in the same pathways and that *nebula* could affect mitochondrial function by modulating ADP/ATP carrier activity [79]. Over the years, the Drosophila *sesB* mutant has become a useful model system for studying the molecular pathology of human diseases associated with AAC dysfunction. A deficiency of ADP/ATP transport has been reported to be implicated in several neuromuscular diseases. These disorders reflect a variety of clinical symptoms ranging from cardiomyopathy and myopathy with lactic acidosis to severe multisystem diseases involving the central nervous system [2]. Recently, a *sesB* mutant defined as *sesB1* aroused great interest [80,81]. In particular, in Drosophila *sesB1*, the point mutation L289F was found to be adjacent to the site of the V298M mutation in the human *SLC25A4*, which was detected in patients with autosomal dominant progressive external ophthalmoplegia (adPEO). The latter is an adult-onset pathology characterized by the paralysis of the muscles responsible for eye movements, although the general symptoms are not limited to the eyes and include exercise intolerance, muscle weakness, hearing deficits and others. Additionally, adPEO is characterized by large-scale mitochondrial DNA (mtDNA) deletions [82]. The bioenergetic and molecular phenotype of *sesB1* fruit fly mutants was well characterized, highlighting significant defects in mitochondrial respiration and oxidative phosphorylation, with a consequent shift towards glycolysis, and fatty acid and amino acid catabolism. It has been proposed that bioenergetic insufficiency manifesting as an ATP deficit and/or altered metabolic or redox homeostasis conferred by decreased AAC activity could promote the onset of the adPEO characteristic symptoms [80].

Currently, four human AAC isoforms have been identified (AAC1-4). They have high amino acid similarities, although the expression patterns differ among them, depending on the cell’s energy requirements and the state of cell differentiation, underlying the critical role of AACs in the maintenance of the energetic fluxes in human cells. As regards the Drosophila homologs, only two AAC genes, known as sesB and ANT2, have been identified based on their homology with other known AACs and on their function in the fruit fly. Mutations affecting sesB in Drosophila were identified in a stress-sensitive strain of flies that are conditionally paralytic in response to mechanical stress. SeSB null alleles are lethal, and knockdown or overexpression also results in developmental lethality, indicating its crucial role in cellular energy metabolism during development. On the contrary, the function of ANT2 has not been fully clarified. However, it is almost exclusively expressed in the testis, similarly to human AAC4, suggesting the presence of a similar energy metabolic pathway in humans and Drosophila. In addition, both Drosophila AACs and human AAC1 and AAC3 are able to induce apoptosis.

### 2.3. The Uncoupling Proteins: CG6492, CG9064 and CG18340 vs. SLC25A7, SLC25A8, SLC25A9, SLC25A14, SLC25A27 and SLC25A30

Mitochondrial uncoupling proteins (UCPs) constitute a subfamily of the SLC25 family allowing the leakage of protons into the mitochondrial matrix, thus dissipating the proton gradient created by the respiratory electron transport chain and uncoupling respiration from ADP phosphorylation [83]. To date, six human isoforms (hUCP1–6) have been identified, encoded by different genes, i.e., *SLC25A7*-*9*, *SLC25A27, SLC25A14* and *SLC25A30* [84]. All the UCPs share amino acid sequences located in the first, second and fourth helices, as well as in the second matrix loop, and named “uncoupling protein (UCP) signatures”; furthermore, the purine nucleotide binding domain (PNBD) is highly homologous [85,86]. These conserved sequences have been analyzed in order to shed light on the evolution of UCPs from a common ancestral gene, suggesting that UCP4 could represent the ancestral UCP from which invertebrate, plant and mammalian UCPs have diverged (firstly, UCP5, and afterward, the others) [26]. Some authors have rebutted this theory, hypothesizing that UCPs diverged very early during evolution (before the divergence of protostomes and deuterostomes) from an ancestral gene into three genetically different clades, probably according to their different functions [27,87]. According to a recent phylogenetic analysis of human UCPs and their closest relatives from other species, three different branches have been detected, including relatives of UCP4–6, UCP1–3 and DIC/OGC proteins. On the first principal branch, UCP5 and UCP6 proteins constitute two close clusters, while UCP4 relatives compose a different cluster [84].

The first identified uncoupling protein was UCP1; the other UCP subfamily members were identified according to their sequence homology with UCP1 [88]. As regards the uncoupling abilities of UCPs, mammalian UCP1 was proven to transport protons into the mitochondrial matrix, significantly uncoupling oxidative phosphorylation from ATP production. The uncoupling abilities of UCP2–5 appear to be moderate; their mild mitochondrial uncoupling could avoid excessive mitochondrial reactive oxygen species (ROS) production, thus reducing cellular oxidative damage [86,89], as recently proved for UCP2 and UCP3 in stroke and ischemia/reperfusion [90]. On the other hand, UCP6 is not able to transport protons but mainly sulfur compounds [84], while UCP2 can catalyze the transport of a dicarboxylate against phosphate plus a proton [91].

Mammalian UCP1, also named thermogenin, is expressed in brown adipose tissue (BAT) and in beige adipocytes [92]. UCP1 can efficiently transport protons across the inner mitochondrial membrane, thus causing a considerable mitochondrial uncoupling and generating heat [88]. This protein is considered essential for adaptive non-shivering thermogenesis in hibernating and newborn mammals [93]. UCP1’s uncoupling activity can be modulated by guanosine diphosphate, which binds and inhibits the carrier, whereas free fatty acids (FFAs) promote its function [94]. A similar modulation has also been shown for UCP2 and UCP3 [83]. The exact UCP1 transport mechanism responsible for proton leak is yet controversial, although it is likely that it might function as a conventional mitochondrial carrier [95,96].

Human UCP2 displays 59% sequence identity with hUCP1, and it is expressed in many tissues including the brain, lung, kidney, spleen, thymus, pancreas and heart, as well as in fast-proliferating cells using aerobic glycolysis, such as cancer, pluripotent stem and immune cells [97]. Human UCP2 represents a mild uncoupling protein and also a metabolite transporter; indeed, it can transport dicarboxylates/inorganic anions, including malate, oxaloacetate, sulfate, phosphate and aspartate in reconstituted systems [91]. In particular, phosphate transport is supposed to be proton-coupled, thus promoting the mitochondrial import of phosphate in exchange for a dicarboxylate or aspartate in energized mitochondria. In this light, UCP2’s biological role could be the exchange of intramitochondrial four-carbon compounds for cytosolic phosphate during cell metabolic reprogramming, when a switch from oxidative phosphorylation towards aerobic glycolysis occurs, as well as during the oxidation of glutamine used for energy metabolism [91]. Furthermore, UCP2 is required in the aspartate–argininosuccinate shunt of the tricarboxylic acid/urea cycle, since this carrier can exchange the cytosolic malate synthesized by cytosolic fumarase from fumarate [98], for intramitochondrial aspartate, which is then utilized in the cytosolic reactions of the urea cycle [99]. The activation of UCP2 can negatively modulate insulin secretion, impairing beta-cell functions [97,100,101,102]. On the other hand, mutations found in hUCP2 are responsible for human congenital hyperinsulinism [103,104]. UCP2 could play a key role in many biological pathways including cell metabolism and proliferation, thus contributing to cancer growth and drug resistance; moreover, polymorphisms of its gene were associated with diabetes and obesity in humans [105,106].

In *D. melanogaster*, the targeted expression of hUCP2 together with mouse UCP1 in adult neurosecretory insulin-like peptide-producing cells (IPCs) resulted in increased uncoupling activity and decreased calcium levels in IPCs. These events led to reduced systemic insulin secretion and signaling, as demonstrated by the sub-cellular localization of the Drosophila homolog of the mammalian forkhead Box O (FoxO) transcription factor, which remained mostly in the nucleus instead of migrating to the cytoplasm [107]. Additionally, moderate hyperglycemia and a decreased expression of insulin-like peptide 3 (ILP3) were found. Interestingly, these metabolic changes extended the lifespan of UCP-expressing fruit flies, also improving their resistance to oxidative stress and starvation. Since K_ATP_ channels were proved to be involved in the mechanism of ILP release, insulin secretion in fruit flies appears to be mediated by the same conserved cascade of intracellular events that also operates in mammalian β-pancreatic cells [107]. In a Drosophila Parkinson’s disease model, hUCP2 expression attenuated brain dopamine depletion, locomotor defects and ATP deficiency by enhancing mitochondrial function without affecting mitochondrial biogenesis. In this regard, an increased complex I activity was found, along with a significantly increased expression of Spargel, i.e., the Drosophila homolog of mammalian peroxisome proliferator-activated receptor gamma coactivator 1-alpha (PGC-1α), which is a transcriptional co-activator involved in energy metabolism and ROS suppression [108]. At the same time, an up-regulation of the mitochondrial transcription factor A (Tfam), a Spargel target gene, was observed, suggesting that hUCP2 expression could activate this pathway. As a consequence, a low endogenous ROS production together with promoted dopaminergic neuronal survival and an extended fruit fly lifespan was observed [109].

Human UCP3 was identified by screening a human skeletal muscle cDNA library, detecting two isoforms with different lengths—i.e., 312 and 275 amino acids—named UCP3L and UCP3S, respectively, and having 57% and 73% sequence identity with the human UCP1 and UCP2, respectively. In humans, UCP3 is abundantly expressed in skeletal muscle and at a low level in the heart, while in rodents, it is expressed in the BAT, heart and skeletal muscle [110]. By the use of planar bilayer membranes and liposomes, UCP3 was shown to transport protons; in particular, the proton transport rate of reconstituted recombinant mouse UCP3 [111] is quite similar to that of hUCP2, but it is fivefold lower than that of hamster UCP1. As already observed for UCP1 and UCP2, FFAs stimulate UCP3 proton transport, even if the underlying mechanism is still unclear. UCP3 activation is higher with increasing FA chain length and unsaturation, with arachidonic acid being the strongest activator [111]. UCP3 proton transport is believed to be activated only under particular conditions, i.e., when FFA blood levels increase; thus, its function could depend on cellular energy metabolism [112]. This transporter is inhibited by purine nucleotides (PNs) and phosphate [111]. The inhibition mechanism by PNs for UCP3 is different from that for UCP1; indeed, the UCP1 maximal inhibition is decreased upon reducing PN phosphorylation, whereas all the PNs can completely inhibit UCP3. Increased FFA levels reduce the effect of all the PNs on UCP1; conversely, FFAs affect only the ATP-mediated inhibition of UCP3 [112]. When hUCP3 was moderately and ubiquitously expressed in *D. melanogaster*, it did not uncouple mitochondria, but its pan-neuronal overexpression led to a significant uncoupling activity in mitochondria isolated from fly heads, causing a drastically shortened lifespan [113]. Similar effects were observed when hUCP3 was specifically overexpressed in median neurosecretory cells (mNSCs), along with increased levels in fly heads of insulin-like peptide 2 (DILP2), which is involved in insulin/insulin-like growth factor (IGF)-like signaling (IIS) [114]. These findings suggest that the secretion of insulin-like peptides could be modulated by the mitochondrial uncoupling rate, affecting lifespan extension.

UCP4 has been exclusively identified in mitochondria isolated from human fetal and adult brain tissues, and it has 29%, 33% and 34% sequence identity with the human UCP1, -2, and -3, respectively, as well as being 40% and 39% identical to the human UCP5 and UCP6, respectively [112]. Further studies have confirmed that it is specifically expressed in the central nervous system, i.e., in highly differentiated neuronal and neurosensory cells needing a high glucose supply and having low proliferation potential [86]. Its expression levels depend on the examined brain region, with the highest protein levels found in the cortex [115]. As found for other UCPs, UCP4 expression in mammalian cells led to a reduction in the mitochondrial membrane potential, thus demonstrating that it could exert a mild uncoupling activity [116]. A neuroprotective role for UCP4 has been hypothesized, since it could reduce ROS generation and modulate the mitochondrial calcium concentration, playing a role in neuronal cell differentiation, apoptosis and neurodegenerative disorders [117].

UCP5 was firstly identified in mice and humans in the central nervous system, where it is mainly expressed, so it is called brain mitochondrial carrier protein-1 (BMCP1) [118]. Three hUCP5 isoforms have been detected: a long isoform including 325 amino acids (UCP5L), a short isoform made of 322 amino acids and containing the deletion of a tripeptide (VSG) at position 23–25 (UCP5S), and an additional isoform including 353 amino acids carrying the deletion and also an insertion of 31 amino acids between the transmembrane domains III and IV, called short insert UCP5 (UCP5SI). Such isoforms have displayed tissue-specific distribution and different abilities to decrease mitochondrial membrane potential [86]. According to the theory that mild mitochondrial uncoupling could represent a protective mechanism, UCP5 was believed to transport protons, thus reducing ROS levels and preserving mitochondria from toxic compounds [119]. UCP5 could also be involved in energy homeostasis, gene regulation and, especially, neuroprotection [116,120]. The hUCP5 protein is 28%, 31%, 31%, 33% and 72% identical to the human UCP1–4 and UCP6, respectively, and it is able to transport many metabolites across mitochondria, including sulfate, thiosulfate, sulfite, l-malate, malonate, maleate, phosphate, oxalate, l-citramalate and d-citramalate, as well as, to a lesser extent, citrate, aspartate and glutamate, as recently found [84]. In the brain, UCP5 could regulate the concentration of important signaling molecules including thiosulfate and sulfite, which are synthesized in mitochondria as H_2_S degradation products; indeed, UCP5 could export them in exchange for cytosolic sulfate, phosphate or a dicarboxylate (malate), thus modulating the high turnover rates that H_2_S has in the brain, affecting vasodilatation and reducing ROS production.

UCP6 has been identified in mouse kidney mitochondria, although it is also considerably expressed in the testes, and it is called kidney mitochondrial carrier protein-1 (KMCP1); remarkably, it does not exert any mitochondrial uncoupling activity in vitro [121]. UCP6 is overexpressed in response to fasting in both proximal and distal tubular cells, when cyclooxygenase-2 protein levels and mitochondrial superoxide dismutase activity are also considerably increased. Furthermore, UCP6 expression is up-regulated during tubular epithelium regeneration after renal injury. Since both of these conditions are characterized by temporary oxidative damage, which is succeeded by increased mitochondrial metabolism and antioxidant defenses, UCP6 is believed to be important in cell redox homeostasis in order to protect cells from oxidative damage when mitochondrial metabolism is increased [121]. The hUCP6 protein is 31%, 35%, 35%, 36% and 72% identical to hUCP1–5, respectively. UCP6 is highly homologous to UCP5, not only from a phylogenetic point of view but also as regards its function, as recently highlighted [84]. Indeed, similarly to hUCP5, hUCP6 was proved to efficiently exchange metabolites, such as sulfate, thiosulfate, sulfite, l-malate, malonate, maleate, phosphate, oxalate, l-citramalate and d-citramalate. On this basis, it is likely that renal hUCP6 could have a functional role similar to that hypothesized for hUCP5 in the brain, because both these tissues have similar H_2_S metabolic pathways [122].

In *D. melanogaster*, four putative UCP isoforms have been identified by using sequence homology and analyzing the “UCP signatures” and PNBD motifs of UCPs from different species [26]. In detail, the *CG6492* gene encodes an isoform considered to be the *D. melanogaster* UCP4 analog, because it has about 52% homology with hUCP4; thus, it has been called *Dm*UCP4A. The *CG18340* and *CG9064* genes encode other close hUCP4 relatives, hence they have been named *Dm*UCP4B and *Dm*UCP4C, respectively. Furthermore, the *CG7314* gene encodes a hUCP5 analog called *Dm*UCP5 (about 51% homologous to hUCP5) [26].

In *D. melanogaster*, UCP4A overexpression was shown to prevent mitochondrial dysfunction and to rescue mutant phenotypes in Parkinson’s disease (PD) models [123]. In this regard, it is known that mutations found in *pink1* and *parkin* genes result in PD [124,125]. Both these genes are involved in quality control pathways triggering autophagy in damaged mitochondria [126,127]. Hence, the loss of function of at least one of them results in the accumulation of dysfunctional mitochondria, especially impairing the function of cells with high energy needs, such as spermatids and muscles in the fruit fly [128,129]. UCP4A overexpression in *pink1* mutants led to restored mitochondrial morphology and activity [123]. In more detail, restored mitochondrial respiration and ATP production, as well as protection from membrane potential loss and decreased ROS levels, were observed in UCP4A-overexpressing *pink1* mutant flies, resulting in decreased oxidative damage and increased resistance to oxidative insults. As regards *pink1* mutant phenotypes, UCP4A overexpression was able to rescue muscle degeneration, locomotor defects, spermatid morphology and male sterility, as well as prolonging dopaminergic neuron survival by protecting them from degeneration. In the same way, *parkin* mutant phenotypes were specifically rescued by UCP4A overexpression, indicating that the mild proton leakage induced by UCP4A was able to attenuate ROS generation during oxidative phosphorylation, thus promoting cell viability under stress conditions [123]. Mild mitochondrial uncoupling was shown to prolong lifespan in different organisms, as highlighted by many studies [130,131,132]. This evidence supports the hypothesis that preserving mitochondrial function during aging through mild uncoupling might extend lifespan, as observed for UCP4A and also for hUCP4, playing a similar protective role in the brain [133]. *Dm*UCP4C was shown to uncouple respiration in *D. melanogaster* larvae, being necessary for larval development at low temperature [134]. In more detail, mitochondria in larval body wall preparations were proved to be uncoupled, since the steady oxygen consumption rate was neither decreased by oligomycin nor increased by the uncoupler carbonylcyanide-p-trifluoromethoxyphenyl hydrazone (FCCP). Additionally, the respiration rate was inhibited by rotenone plus antimycin A; additionally, iodoacetate and 2-deoxyglucose (inhibitors of glycolysis) were able to significantly decrease the steady oxygen consumption rate. All these findings indicated that only glycolysis produced the ATP necessary for larval survival and early development [134]. The knockdown of *Dm*UCP4C, but not that of *Dm*UCP4A or *Dm*UCP4B [135], led to mitochondrial recoupling, along with larval lethality at 15 °C (but not at 23 °C). In order to understand whether uncoupled respiration was linked to heat generation, larvae were reared at 23 °C, shifted to 14 °C for 1 h, and returned to 23 °C. Notably, the wild-type larva temperature was 0.5 °C higher than that of the medium, whereas no thermal gradient was observed for Ucp4C-modulated larvae, indicating that wild-type mitochondria generated heat through *Dm*UCP4C uncoupling activity, which was strictly required for larval survival and development at low temperature [134]. Interestingly, the uncoupled respiration of larval mitochondria was enhanced by palmitate and inhibited by guanosine diphosphate, as already observed for mammalian UCP1-3. Moreover, among *Dm*UCPs, *Dm*UCP4C exhibited the highest homology to hUCP1 [134], suggesting that *Dm*UCP4C could exert a thermogenic function, at least in *D. melanogaster*. A loss of circadian regulation was proved to significantly extend *D. melanogaster* lifespan by upregulating *Dm*UCP4C’s uncoupling activity in the intestine [136]. In detail, the functional disruption of at least one of the genes *period* (*per*) and *timeless* (*tim*)—encoding the circadian transcriptional repressors Per and Tim, respectively [137]—surprisingly extended lifespan in Drosophila males, without affecting canonical pathways responsible for longevity, such as autophagy, insulin and TOR signaling [138,139]. Additionally, this phenomenon was independent of dietary restriction [140]. In particular, *per* mutants were deeply investigated, and they were phenotypically hyperphagic and leaner with respect to the wild-type controls. Moreover, they had increased respiration rates and mitochondrial uncoupling, along with decreased mitochondrial membrane potentials and faster recovery from cold shock than control flies. Notably, *DmUcp4B*/*C* expression was proved to be essential for maintaining metabolic phenotype, as well as for longevity [136]. Furthermore, *Dm*UCP4C overexpression resulted in flies miming the phenotype already observed for *per* mutants, including an extended lifespan. The intestinal circadian clock appeared to play a critical role, since the loss of intestinal Per was essential for the prolonging of lifespan, and this effect was mediated by *Dm*UCP4C. In detail, intestinal *Ucp4C* mRNA levels were proved to be low during the day and high at night in control flies, whereas in *per* mutants, they were constitutively elevated both night and day, indicating that mitochondrial uncoupling is circadian-regulated at least in the intestine of *D. melanogaster*. Hence, the mitochondria of *per* mutants were uncoupled both night and day, and this continuous uncoupling reduced intestinal ROS production, thus delaying aging-related intestinal barrier dysfunction and maintaining gut barrier function. In addition, decreased intestinal ROS levels suppressed stem cell overproliferation in both aging and tumorigenesis, resulting in an extended lifespan [136]. These findings support the hypothesis that *Dm*UCP4C can exert a protective role similar to the roles observed for UCP4A in fruit fly neurons and for hUCP4 in the brain, since all these proteins are able to mild uncouple mitochondria and to reduce ROS generation, thus prolonging lifespan.

*Dm*UCP5 was heterologously expressed in a yeast system, in which it increased the respiration rate and decreased the mitochondrial membrane potential in the presence of oligomycin, thus suggesting that *Dm*UCP5 is endowed with uncoupling activity [141]. Moreover, the mitochondrial respiration rate was enhanced by fatty acids and inhibited by guanosine diphosphate, indicating that the uncoupling activity of *Dm*UCP5 is regulated in the same way as hUCP1-3 and UCP4C [134]. *Dm*UCP5 is expressed in all the fruit fly developmental stages, although its level is significantly increased in adult flies, especially in adult heads that are mainly constituted by brain [141], thus highlighting an expression pattern similar to that found for hUCP5. *Dm*UCP5 was also hypothesized to play a role in *D. melanogaster* metabolism and aging [142]. In this regard, *Dm*UCP5 knockout (UCP5KO) flies developed normally and did not present morphological alterations, but UCP5KO females were little fertile, laying a lower number of eggs than controls. Unexpectedly, UCP5KO flies were highly starvation-sensitive, having shorter lifespans than controls. Additionally, they lived longer on low-caloric feeding (but not on high-caloric feeding) and gained less weight on high-caloric feeding than controls [142]. Mitochondria isolated from the heads and thoraces of UCP5KO flies showed normal uncoupling rates measured in the presence of oligomycin, as well as normal ATP contents when compared to the control flies, implying that imbalanced metabolic homeostasis due to *Dm*UCP5 loss could arise from a limited population of cells, likely unable to affect overall ATP production. In addition, UCP5KO flies exhibited lower glucose and trehalose levels than controls, also in starved conditions. Both types of flies consumed their glycogen supplies at similar rates during starvation, whereas UCP5KO flies used their triglyceride supplies at a faster rate than control flies during starvation. This evidence suggested that UCP5KO flies were hypoglycemic and had to mobilize their triglyceride storages faster than controls to preserve their basal metabolism. Nevertheless, when starvation was prolonged, the UCP5KO flies depleted their reserves more rapidly and died faster than controls [142]. Consistently with these findings, and considering that ectopic neuronal *Dm*UCP5 expression was able restore starvation resistance in UCP5KO flies, this carrier could be involved in the control of metabolic homeostasis at the brain level, by uncoupling mitochondria in brain neurosecretory cells and affecting their secretion of adipokinetic hormone (the fruit fly equivalent of glucagone) and insulin-like peptides [143]. In this light, *Dm*UCP5 could interfere with hormonal metabolic homeostasis, as observed for mammalian UCP2 [103,104]. On the other hand, considering that hUCP5 was proved to transport many metabolic intermediates [84] and that it is 53% identical to *Dm*UCP5 [141], the latter could also transport across the inner mitochondrial membrane some compounds able to regulate metabolic homeostasis. This hypothesis could be confirmed by carrying out *Dm*UCP5 kinetic characterization in reconstituted systems.

On the basis of sequence homology, two kinds of UCP related to humans have been detected in Drosophila—*Dm*UCP4, with three different transcripts (*Dm*UCP4A, *Dm*UCP4B and *Dm*UCP4C), and *Dm*UCP5. Despite the homology, it is difficult to ascribe the Drosophila UCPs simply as the homologs of hUCP4 and hUCP5, since they present characteristics also belonging to the other human UCPs. All *Dm*UCPs as hUCP2-5 show a mild uncoupling activity that extends lifespan as observed for hUCP4. *Dm*UCP4A exerts its function in cells with high energy needs, such as spermatids and muscles; *Dm*UCP4C is expressed in larvae, whereas its human homolog has been identified in human fetuses and adult brains. Interestingly, the mitochondrial respiration rates of *Dm*UCP4C and *Dm*UCP5 were enhanced by palmitate and inhibited by guanosine diphosphate, indicating that the uncoupling activity is regulated in the same way as that of hUCP1-3. Remarkably, *Dm*UCP4C is the only one able to generate heat, suggesting that isoforms, such as hUCP1, could exert a thermogenic function. *Dm*UCP5 is expressed especially in the adult brain, highlighting an expression pattern similar to that found for hUCP5. Furthermore, as observed for mammalian UCP2, it has been hypothesized that *Dm*UCP5 could interfere with hormonal metabolic homeostasis.

### 2.4. The Dicarboxylate Carriers: CG8790, CG4323, CG11196 and CG18363 vs. SLC25A10

In mammals, only one gene named *SLC25A10* encodes the dicarboxylate carrier (DIC), which was firstly isolated from rat liver mitochondria and tested for its transport activity [2]. DIC’s full characterization was achieved in rats, with reconstituted proteoliposomes, either with the protein purified from mitochondria or using a heterologous expression bacterial system [2]. Rat DIC acts as an electroneutral obligatory antiporter, transporting dicarboxylate ions (l-malate, malonate and succinate), inorganic phosphate and some organosulfur metabolites including sulfate, thiosulfate and sulfite [2]. A debated question is whether DIC may transport glutathione [144,145,146]. Mammalian DIC is essential in many metabolic processes, since it can sustain gluconeogenesis by transporting intramitochondrial malate in exchange for cytosolic phosphate. It also plays a critical role in urea synthesis by importing the malate synthesized from fumarate by argininosuccinate lyase, as well as being able to affect sulfur metabolism, allowing the export of intramitochondrial sulfite or sulfate in exchange for cytosolic cysteinsulfinate, thiosulfate and hydrogen sulfide [1,2,147]. Recently, *SLC25A10* mutations were proved to be responsible for a serious disease causing epileptic encephalopathy and mitochondrial DNA depletion in skeletal muscle [2,148]. Furthermore, DIC was found to be involved in the onset of diabetic nephropathy and cancer progression [144,149,150]. According to all these multifaceted roles, human DIC (hDIC) is expressed in many tissues, such as the liver, kidney, brain, heart, pancreas, lung and adipose tissue [144].

When the *D. melanogaster* genome was screened using the hDIC sequence, a subfamily of four genes (*CG8790*, *CG4323*, *CG11196* and *CG18363*) was detected, encoding four putative homologous carriers named *Dm*Dic1p, *Dm*Dic2p, *Dm*Dic3p and *Dm*Dic4p, respectively, based on their homology with hDIC [24]. In detail, pairwise alignments of these proteins with hDIC highlighted 57%, 46%, 45% and 35% identity for *Dm*Dic1p, *Dm*Dic2p, *Dm*Dic3p and *Dm*Dic4p, respectively. All the *Dm*Dic proteins (*Dm*Dicps) have sequence features shared by the other mammal SLC25 family members [1]. Expression analysis during development performed by semi-quantitative RT-PCR revealed that *Dm*DIC1 is expressed at high levels in all the considered developmental stages (embryos, larvae, pupae and adults) and *Dm*DIC2 remained undetected, whereas *Dm*DIC3 and *Dm*DIC4 are highly expressed only in the pupal stage. Recombinant *Dm*Dicps were overexpressed in bacterial cells as inclusion bodies (with the exception of *Dm*Dic2p, whose gene amplification repeatedly failed), then they were purified, reconstituted into liposomes and tested for their transport activity. *Dm*Dic1p works as a strict antiporter, exchanging l-malate or phosphate for l-malate, malonate, phosphate, maleate and, to a lesser extent, succinate, sulfate, thiosulfate and oxalacetate [24]. The l-malate/phosphate exchange reaction is strongly inhibited by benzylmalonate or butylmalonate (i.e., substrate analogs) and other inhibitors commonly used for many SLC25 family members, including bathophenanthroline (BAT) and pyridoxal 5′-phosphate (PLP) (98%), whereas some inhibitors used for other characterized transport proteins have proved to be ineffective, including CATR [151] and 1,2,3-BTA [3]. The *Dm*Dic1p Km values for l-malate and phosphate are about 0.81 and 2.35 mM, respectively [24], while its Vmax value is about 64 μmol/min×mg protein for both phosphate/phosphate and l-malate/l-malate exchanges, and such values are virtually independent of the kind of substrate, as already observed for rat Dic proteins [1]. The *Dm*Dic1p Km values for l-malate and phosphate are similar to those determined for the recombinant rat carrier, whose values are 0.78 and 1.77 mM, respectively [1]. Interestingly, functionally reconstituted *Dm*Dic3p transport phosphate in strict antiport only for sulfate, phosphate and thiosulfate but not for l-malate, at variance with *Dm*Dic1p and rat DIC [2,24]. *Dm*Dic3sp’s transport activity is strongly inhibited by BAT, PLP and mersalyl, whereas inhibitors (CATR or 1,2,3-BTA) used for other SLC25 family members do not affect its transport function. In homo-exchange experiments, the *Dm*Dic3p Km value for phosphate and Vmax value have been estimated to be about 3.2 mM and 253 μmol/min×mg protein, respectively [24]. Unexpectedly, *Dm*Dic4p was unable to catalyze either homo- or hetero-exchanges, after its reconstitution into liposomes [24].

The differences between *Dm*Dic1p, *Dm*Dic3p, *Dm*Dic4p and mammalian DIC were further investigated and explained by structural analysis, comparing their protein sequences with those of ketoacid carriers and the bovine ADP/ATP carrier, whose structure was solved by X-ray crystallography [152]. Such data have highlighted that all the *Dm*Dic carriers can be better classified within ketoacid than within phosphate carriers. Moreover, homology modeling studies of *Dm*Dicps and the 2-oxoglutarate carrier (OGC) have shed light on their transport features [24]. Bovine OGC was chosen since it represents the most well-characterized SLC25 family member, as it was the subject of systematic cysteine-scanning mutagenesis studies aiming to identify functionally important amino acid residues [6,153,154,155]. In this regard, it is noteworthy that all the SLC25 family members share three conserved contact points (CP1, CP2 and CP3) representing the common substrate binding site, and they contain amino acid residues essential for transport activity [6]. In *Dm*Dic1p, a strict conservation of all the three contact points has been highlighted, together with the conservation of their flanking residues explaining its typical activity. From a structural point of view, *Dm*Dic3p’s peculiar activity and inhibitor sensitivity could arise from mutations found in regions able to affect carrier activity, which flank the substrate binding site. They could cause a higher steric hindrance degree in *Dm*Dic3p with respect to what happens in *Dm*Dic1p, promoting the access of phosphate to the substrate binding site and hampering that of l-malate. Furthermore, this hypothesis is in agreement with previous findings indicating that two separate binding sites could exist in rat DIC; one could be specific for phosphate (or phenylphosphate), and the other could specifically bind a dicarboxylate (or butylmalonate); these sites could be bound by their specific substrates in the absence of mutual interference [156]. As regards *Dm*Dic4p, its inability to transport dicarboxylates or phosphate could arise from a loss of some conserved residues located on CP1 and CP2. Its gene could be derived from a duplication of the *Dm*DIC ancestral gene, but the lack of selective pressure upon this gene copy would have allowed mutations impairing *Dm*Dic4p’s transport function [24].

In conclusion, on the basis of functional and structural data, *Dm*Dic1p could be the *D. melanogaster* homolog of mitochondrial mammalian DIC, and in this insect, it could affect sulfur metabolism and sustain gluconeogenesis from pyruvate but not urea synthesis, because this organism is uricotelic and uses an arginase-independent pathway [157]. On the other hand, *Dm*Dic3p could represent an atypical dicarboxylate carrier because of its ability to transport phosphate (instead of l-malate) for phosphate and also because of its peculiar inhibition sensitivity with respect to that of *Dm*Dic1p and of other characterized dicarboxylate carriers [1,2]. It is likely that *Dm*Dic3p does not represent the main Drosophila phosphate carrier necessary for the basal synthesis of mitochondrial ATP for many reasons, which are mainly related to its limited substrate specificity, low affinity for phosphate, exclusive expression in the pupal stage and low percentage similarity with human phosphate carriers (17.4% and 17.8%) [1]. *Dm*Dic3p represents a *Dm*Dic1p paralog, and it could fulfill the increased energy needs during metamorphosis in the pupal stage, when it is solely expressed [24].

### 2.5. The Glutamate Carrier: CG18347 and CG12201 vs. SLC25A18 and SLC25A22

Glutamate is a multifunctional amino acid implicated in numerous metabolic and signaling functions. It is an important neurotransmitter and intracellular messenger, a metabolic intermediate of the Krebs cycle and a key member of ammonia metabolism. This essential amino acid is closely associated with mitochondrial metabolism, which is tightly controlled by the activities of mitochondrial enzymes and carrier proteins [99].

Mitochondrial glutamate transport has been the object of many studies, which have led to the identification and characterization of two isoforms in humans, encoded by the *SLC25A22* and *SLC25A18* genes and named GC1 and GC2, respectively [158]. Both are ubiquitous, but *SLC25A22* is expressed at higher amounts in all the tissues, and it is particularly abundant in the liver and pancreas. Glutamate carriers (GCs) directly participate in cellular metabolic processes, such as amino acid degradation, nitrogen metabolism and the urea cycle, by catalyzing the import of glutamate together with a proton from the cytosol to the mitochondrial matrix. Then, glutamate transport is strongly influenced by the pH gradient generated by the respiratory chain, and it is promoted in energized mitochondria [158]. However, under certain conditions, GCs can operate in the reverse direction, limiting glutamate accumulation within mitochondria. This was confirmed by studies investigating the role of GC1 in the control of glucose-stimulated insulin secretion. Indeed, it was shown that GC1 silencing in insulin-secreting β-cells inhibited the secretory response to high glucose due to a reduction in glutamate export from mitochondria [159].

The Drosophila *CG18347* and *CG12201* genes have been identified as candidates for glutamate carriers in *D. melanogaster* after screening the Flybase web server with the sequences of human GCs [160]. The *CG18347* gene encodes a protein indicated as *Dm*GC1p. It is abundantly expressed across various Drosophila developmental stages and in different adult tissues, particularly in the abdomens of adult flies, where the fat bodies mainly accumulate, demonstrating that in *D. melanogaster*, this isoform has an essential role in mitochondrial glutamate metabolism. In contrast, the expression profile of the *CG12201* gene, encoding the protein named *Dm*GC2p, exhibits male-specific expression confined to the testis [160]. Both *Dm*GC1p and *Dm*GC2p catalyze the transport of l-glutamate either with H^+^ or in exchange for OH^-^, and do not transport structurally related compounds, such as aspartate, glutamine and asparagine, as do human GCs. However, both the Drosophila and human isoforms markedly differ in their kinetic parameters. *Dm*GC1p has a low Km value for glutamate (about 0.4 mM), whereas the Km value of *Dm*GC2p is very high (about 3mM); additionally, *Dm*GC2p is two times less active than *Dm*GC1p. On the other hand, GC1 has a very high Km value for glutamate (4–5 mM), whereas the Km value of GC2 is lower (about 0.2 mM), and the Vmax value of GC1 is higher than that of GC2 [158,160].

Comparative analysis of the *Dm*GC1 and *Dm*GC2 coding sequences has revealed that they share with human GCs the specific amino acid triplets characterizing the GC subfamily. Only a difference in *Dm*GC2p triplet 22 has been detected, where a serine is present instead of an alanine residue, characterizing the same position in *Dm*GC1p and in human GCs [11,160]. Another important difference has been found corresponding to the leucine residue of *Dm*GC2p triplet 81, probably involved in the formation of the substrate-binding site, based on a comparison with its homologs, in which an isoleucine residue was detected [11,160]. It is likely that this latter residue might be responsible for the slightly different interactions of glutamate within the *Dm*GC1p cavity, at the level of the proposed binding site. In this regard, residues of triplet 81 (in particular, arginine–aspartic acid dipeptides), together with those of triplets 80 and 77 (in particular, the arginine residue), appear to be involved in direct interactions with the glutamate in the substrate-binding site of *Dm*GCs [11,160]. In addition, two residues found mutated in GC1 deficiency (P206L and G236W) [2] appeared to be conserved both in *Dm*GC1p (P205 and G237) and *Dm*GC2p (P204 and G236), suggesting that they might be involved in the functional activity of both human GCs and Drosophila orthologs.

A significant difference between the two *Dm*GC genes resides in the occurrence, in the *CG18347* gene, of the NRG element, a palindromic 8-bp motif present in all the nuclear insect OXPHOS genes [30,161]. Furthermore, single or multiple NRG elements were detected in the non-coding regions of the GC genes of several other Drosophilidae and non-Drosophilidae arthropod species. This evidence confirms the key role played by *Dm*GC1 in the regulation and maintenance of energy metabolism, in contrast with that of *Dm*GC2, which could be a gene in evolution with potential novel functions. Kinetic parameters, tissue distribution, and evolutionary and modeling studies have highlighted deep differences in the biological functions of the two Drosophila isoforms: *Dm*GC1p might be responsible for the basic functions of amino acid catabolism and uricogenesis, whereas *Dm*GC2p activity could be required to satisfy the higher energy demand associated with specialized spermatozoan functions. Similarly, different physiological roles for hGCs were reported, since GC2 meets the basic requirements of all the tissues, whereas GC1 could be necessary for specific metabolic functions and conditions.

### 2.6. The Thiamine Pyrophosphate Carrier: CG6608 and CG2857 vs. SLC25A19

Several cofactors are essential for the functioning of important metabolic processes occurring in mitochondria. Thiamine pyrophosphate (ThPP) is a fundamental coenzyme of various cytosolic and mitochondrial reactions, including carbohydrate, lipid and branched-chain amino acid metabolism. Given its great biological relevance, the transport of ThPP across the inner mitochondrial membrane has been extensively studied in several organisms. In humans, only recently has a ThPP mitochondrial transporter (TPC) been identified through homology searching by using the *Saccharomyces cerevisiae* ThPP carrier (Tpc1p) sequence [162], and it has been characterized by transport assays using the recombinant reconstituted protein [163]. Human TPC is encoded by the *SLC25A19* gene, and previously, it was believed to be a mitochondrial deoxyribonucleotide carrier, on the basis that it clustered with the ADP/ATP carrier (AAC) and catalyzed an exchange reaction between nucleotides and deoxynucleotides [164]. Subsequently, ThPP and thiamine monophosphate (ThMP) were also found to be very efficiently transported by TPC [163], as observed for its yeast homolog Tpc1p [162]. In mice, the loss of *SLC25A19* function did not alter the mitochondrial nucleotide pool, not affecting mitochondrial function, DNA synthesis or integrity, thus confirming that the transport of deoxynucleotides into the mitochondrial matrix is not the primary role of the *SLC25A19* gene product. On the other hand, in knockout mice, reduced mitochondrial ThPP content along with alterations in mitochondrial ThPP-dependent enzyme activities was observed [163]. In more detail, these mutants presented not only metabolic alterations but also central nervous system defects, which were similar to those observed in patients affected by Amish lethal microcephaly (MCPHA), a syndrome associated with a *SLC25A19* loss-of-function mutation [165].

The *D. melanogaster* thiamine pyrophosphate carrier was identified based on sequence similarity searching by using the protein sequence of the human TPC [166]. Three putative transcripts corresponding to the *D. melanogaster* genes *CG6608* and *CG2857* have been identified. The *CG6608* gene encodes two transcripts (*CG6608-RA* and *CG6608-RB*), resulting in the production of a single protein, named *Dm*Tpc1p, and they are expressed during all the *D. melanogaster* developmental stages (embryos, larvae, pupae and adults). On the contrary, no transcripts were detected for the intronless *CG2857* gene in any developmental stage, indicating that it could be a paralogous gene produced by the retrotransposition of the pre-existing “parent” *CG6608* gene [166]. *Dm*Tpc1p functional and kinetic characterization has revealed that the recombinant protein reconstituted in liposomes mostly catalyzes the transport of ThPP and, to a lesser extent, pyrophosphate, ADP and ATP, which are transported with a higher efficiency than the remaining nucleoside (deoxy)diphosphates and (deoxy)triphosphates. No significant exchange activity has been found using thiamine, ThMP, nucleoside (deoxy)monophosphates, nucleosides, purines or pyrimidines [166]. The biochemical properties of the recombinant reconstituted *Dm*Tpc1p are only partially different from those of the human TPC protein. In this regard, *Dm*Tpc1p has a higher affinity for ThPP with respect to TPC, and probably effective counter-exchange substrates for ThPP are ATP (NTPs), ADP (NDPs) and PPi, the Drosophila isoform being unable to transport ThMP. Conversely, TPC catalyzes an obligatory counter-exchange, preferentially of cytosolic ThPP for mitochondrial ThMP [164]. Hence, because ThPP is produced in the cytosol, the main role of *Dm*Tpc1p could be to catalyze the uptake of ThPP into the mitochondrial matrix, where it is fundamental for mitochondrial metabolism, in exchange for internal ATP, which is generated by oxidative phosphorylation [166].

Recently, the ability of *Dm*Tpc1 to transport cisplatinum-bonded nucleotides within the mitochondria was investigated, thus suggesting a possible direct effect of platinated complexes on these organelles [167]. In this regard, mitochondrial uptake represents a necessary prerequisite for the incorporation of platinated bases into the mitochondrial DNA (mtDNA), with the consequent inhibition of replication and transcription. These events could emphasize some aspects of cisplatin’s antitumor activity. Firstly, the uptake of the model platinum (Pt) purine complexes [Pt(dien)(N7-5′-dGTP)] and cis-[Pt(NH3)2(py)(N7-5′-dGTP)] was studied by using phospholipid vesicles (liposomes) reconstituted with the recombinant *Dm*Tpc1 [167]. The uptake of both complexes was assessed by measuring the amount of Pt incorporated into proteoliposomes by ICP-AES (inductively coupled plasma atomic emission spectroscopy). On this basis, it is likely that *Dm*Tpc1 catalyzes a specific transport of platinated purines, although this carrier has a lower affinity for both model complexes with respect to that for dNTPs. The uptake of model platinated nucleotides in freshly isolated rat liver mitochondria has also been investigated, along with their possible insertion into the newly synthesized mtDNA [168]. Time course experiments have provided evidence for a rapid and highly selective uptake of Pt purine complexes, specifically catalyzed by TPC, into freshly isolated rat liver mitochondria, suggesting that TPC could be directly involved in the cytotoxicity of nucleoside-analog-based drugs. Moreover, in organello mtDNA synthesis assays have revealed that these complexes are incorporated into the mtDNA by DNA polymerase γ activity [168]. Remarkably, for the first time, a mitochondrial carrier was found to be directly involved in the transport of metalated purines related to cisplatin’s mechanism of action.

### 2.7. The Carnitine/Acylcarnitine Carrier: CG3057 and CG3476 vs. SLC25A20

In mammals, the *SLC25A20* gene encodes the carnitine/acylcarnitine carrier (CAC), catalyzing the electroneutral 1:1 exchange of cytosolic acylcarnitine esters for intramitochondrial free carnitine [169]. This carrier, together with carnitine palmitoyltransferase 1 (CPT1) and 2 (CPT2), constitutes the carnitine shuttle, which allows the transport into the mitochondrial matrix of long-chain fatty acids to be oxidized by beta-oxidation. This latter represents the main source of energy production during exercise in cardiac and skeletal muscle, as well as in extended fasting [170]. CAC purified from rat liver mitochondria revealed similar substrate specificity and inhibitor sensitivity either in intact isolated mitochondria or in reconstituted proteoliposomes [169]. Further functional information came from the functional reconstitution of recombinant CAC expressed in *E. coli* [169] or in *S. cerevisiae* [171]. Transported fatty acyl chains have a length between 2 and 18 carbon atoms, even if mammalian CAC transports long fatty acyl chains with higher affinity, whereas the yeast carrier prefers short fatty acyl chains [172]. Differently from many other mitochondrial carriers that share a sequential mechanism of reaction, CAC is characterized by a ping-pong transport mechanism, allowing it to catalyze exchange as well as uniport reactions, albeit to very small extent in the latter case. CAC’s uniport reaction is believed to be important, since it allows the net import of carnitine into the mitochondria, in order to balance its mitochondrial matrix level with that of cytosol [172]. CAC activity is strongly inhibited by specific thiol group reagents including *N*-ethylmaleimide (NEM) and mersalyl, as well as by substrate analogs such as sulfobetaines and acyl-d-carnitine [169,172,173]. CAC is highly expressed in the liver, heart, and skeletal muscle and, to a minor extent, in the brain, placenta, kidney, pancreas and lung [169]. Mutations found in the human *SLC25A20* gene are responsible for CAC deficiency (*OMIM 212138*), which can manifest in two phenotypes depending on residual CAC activity: a neonatal-onset severe type and an infancy-onset milder form. Both phenotypes can result in metabolic abnormalities, including increased plasma concentrations of long-chain acyl-carnitines, ammonia, transaminases and creatine kinase, whereas the free carnitine level is decreased; moreover, hypoglycemia, hypoketosis and dicarboxylic aciduria are present. Heart, skeletal muscle and liver damage can also be present; moreover, patients may experience vomiting, weakness, hypotonia, respiratory distress, neurological dysfunctions and seizures. Fasting or illness can potentially lead to hypoglycemia, neurological damage, coma and death [172].

A *SLC25A20* homologous gene was identified in *D. melanogaster*, and it was proved to be essential for tissue-specific morphogenetic mechanisms including tracheal gas-filling and the expansion of the wings after eclosion [174]. This homolog was detected during a screen aimed to identify genes controlling wing morphogenesis, which represents a useful strategy for investigating genetic events governing larval development, cell differentiation and morphogenesis during pupal metamorphosis [175,176]. By P-element insertional mutagenesis performed using a *P-lacW* transposon, mutations were found in a gene able to deeply affect fruit fly survival. In particular, homozygous first-instar larvae displayed about 70% lethality and were unable to move. Such larvae had a tracheal tree completely formed but partly filled with fluid. Conversely, wild-type larvae were very mobile and had a distended tracheal tree filled with gas. According to this evidence, the mutated homologous gene responsible for this phenotype was named *congested-like tracheae* (*colt*). Surviving insects could complete development, but females were found to be totally sterile, whereas about a quarter of males were able to breed. All adult survivors presented a significant decrease in wing size, especially in their width, with a remarkable loss of venation [174]. The *colt* gene was identified by mapping the position of the P-element insertion by in situ hybridization; it was found in the cytological area 23A5–23B1-2. Interestingly, the observed alterations could be reverted upon the mobilization of the *P-lacW* transposon, thus confirming that *colt* phenotype arose from this P-element insertion [170]. The *colt* gene originates a transcript expressed during each developmental stage. Its corresponding protein has an apparent molecular mass of 33kDa and is considered a member of the SLC25 mitochondrial carrier family because of its structural organization, which contains the main signature motifs shared by all the SLC25 family members. In particular, the Colt amino acid sequence is highly identical (49.5%) to that of the *Caenorhabditis elegans* DIF-1 protein, which is a mitochondrial carrier homolog important during embryonic tissue differentiation [177]. Colt deficiency was found to be critical during eclosion time, when it affected the epithelial morphogenesis of the wings, as well as during larval development, since the absorption of tracheal fluid and its substitution by gas were impaired, thus causing a collapse of the tracheal tree and high lethality. Considering that the absorption of fluid through the tracheal epithelium is believed to be an active process requiring energy, it was supposed that Colt was involved in energy production, even if its activity was not tested [174]. A subsequent study highlighted that Colt is a mitochondrial carnitine/acylcarnitine carrier [178]. In particular, a sequence alignment of the CAC orthologs of *H. sapiens* and *S. cerevisiae* YOR100c, along with Drosophila Colt and *C. elegans* DIF-1, revealed that both of these latter proteins are CAC homologs. In order to confirm their function, their genes were individually used to rescue a double-deletion strain of *S*. *cerevisiae* by functional complementation assays [2,179]. In yeast, fatty acid beta-oxidation occurs in peroxisomes, producing acetyl CoA as a final product; the latter must exit the peroxisomes and can enter mitochondria either via the glyoxylate cycle, in which peroxisomal citrate synthase (CIT2) is the key enzyme, or via CAC. When the CIT2 and CAC genes are both deleted in yeast, acetyl CoA cannot enter the mitochondrial matrix, hence it cannot originate CO_2_. The independent expression of *dif-1* or *colt* in the yeast *Δcac*/*Δcit2* double mutant was able to re-establish CO_2_ generation, thus confirming that in *D. melanogaster*, Colt is an ortholog of CAC, and it is believed to play a key role in energy production during embryonic development [178]. More recently, Colt was hypothesized to be involved in lipid metabolism alterations in a *D. melanogaster* model of amyotrophic lateral sclerosis (ALS) [180]. In fruit fly larvae, the overexpression in motor neurons or in glial cells of the human TAR DNA binding protein 43 (TDP-43), either the wild-type TDP-43 (TDP-43^WT^) or a disease-associated mutant (TDP-43^G298S^), resulted in locomotor defects, synaptic abnormalities at the neuromuscular junction and a reduced life span [181,182]. The metabolomic profiling of larval cells expressing TDP-43^WT^ or TDP-43^G298S^ in motor neurons revealed a remarkable increase in carnitine-conjugated long-chain fatty acids and a relevant decrease in acetyl-carnitine, carnitine and beta-hydroxybutyrate levels, indicating reduced mitochondrial lipid beta-oxidation [180]. The overexpression of each of these proteins modulated the transcriptional levels of all the carnitine shuttle components (i.e., CPT1, CPT2 and Colt), suggesting a deficiency in this transport system, which is also essential for importing long-chain fatty acids into mitochondria in *D. melanogaster*. On this basis, TDP-43-expressing larvae were fed using foods supplemented with beta-hydroxybutyrate or with medium-chain fatty acids (6–12 carbons and fewer), which can cross the inner mitochondrial membrane independently of this shuttle [183]. Dietary supplementation relieved locomotor dysfunction related to TDP-43 in motor neurons, suggesting that carnitine shuttle impairment might be involved in TDP-43-induced toxicity in ALS [180].

Surprisingly, another gene homologous to *colt* was recently found in *D. melanogaster* by sequence comparison using the yeast mitochondrial Mg^2+^ exporter Mme1 as the query [184]. In detail, five *D. melanogaster* genes having high similarity to the yeast MME1 were detected, including *colt* and *CG3476*; the latter showed 30% identity and 45% similarity to the yeast MME1 and acted as a mitochondrial Mg^2+^ exporter, as demonstrated by yeast functional complementation [184]. *CG3476* was heterologously expressed in wild-type yeast, causing group II RNA splicing defects and growth deficiency on respiratory medium, along with an appreciable reduction of intramitochondrial Mg^2+^ levels, without affecting those of Zn^2+^, thus confirming its ability to export Mg^2+^. Hence, it was considered a Drosophila ortholog of yeast Mme1 and was named dMME1 [184]. Similarly, when dMME1 was heterologously expressed in *E. coli*, it localized to the plasma membrane and proved to act as a Mg^2+^ exporter, since dMME1 specifically decreased Mg^2+^ cellular levels. According to sequence similarity, this transporter was shown to belong to the SLC25 mitochondrial carrier family, and its localization to the mitochondrial membrane was proved in CHO cells by fluorescence microscopy [184]. The function of dMME1 was also investigated in vivo, creating dMme1 knock-down by RNAi. The resulting dMME1-RNAi fruit flies showed an increased intramitochondrial content of Mg^2+^, together with a decreased post-mitochondrial Mg^2+^ level, whereas Zn^2+^ and Ca^2+^ contents were unaffected. Interestingly, such silencing significantly shortened the lifespan of the dMME1-RNAi fruit flies, suggesting that intramitochondrial Mg^2+^ accumulation could influence fruit fly longevity. In order to confirm this hypothesis, after eclosion, dMME1-RNAi adult fruit flies were shifted to a Mg^2+^-deficient synthetic diet for two weeks, and, as expected, their survival defects were rescued; additionally, their intramitochondrial Mg^2+^ levels became similar to those of the controls. Notably, when dMme1 was overexpressed in fruit flies, a shortened lifespan and a reduced intramitochondrial Mg^2+^ level together with a corresponding increased post-mitochondrial Mg^2+^content were observed. These defects could be fully rescued by culturing dMME1-overexpressing fruit flies for two weeks on the normal synthetic diet supplemented with Mg^2+^. It is noteworthy that both variations of dMme1 expression (i.e., knock-down or overexpression) led to a little variation (about 10%) in the intramitochondrial Mg^2+^ contents in the corresponding directions, suggesting that in *D. melanogaster*, a fine regulation of intramitochondrial Mg^2+^ concentration is required to ensure an optimal lifespan [184]. Based on the current state of knowledge, both proteins encoded by *colt* and *CG3476* should be heterologously overexpressed, purified and studied using bioinformatic and homology modeling approaches. Furthermore, their functional role should be ascertained by a full kinetic characterization in reconstituted systems such as proteoliposomes. Those approaches would be useful for defining their substrate specificity, allowing an understanding of how mitochondrial carriers having a significant sequence similarity can transport substrates so differently from a structural point of view (i.e., acylcarnitine esters vs. Mg^2+^). Those studies would also allow the clarification of the biological meaning of such transports in *D. melanogaster* and how they could possibly be related, paving the way for the discovery of new probable biochemical pathways in humans.

### 2.8. The Mitoferrin: CG4963 vs. SLC25A28 and SLC25A37

Iron is an essential cofactor in many fundamental biological processes, including energy metabolism, cellular respiration and DNA synthesis. However, iron is also potentially toxic because it promotes the generation and propagation of ROS. Mitochondria are well known to play a pivotal role in iron metabolism, being the main site of iron utilization and accumulation. Therefore, a fine regulation of iron transport in mitochondria is essential for cellular iron balance [185,186]. Today, there are several pieces of evidence of the existence of regulatory processes coupling cytosolic and mitochondrial iron metabolism, including iron uptake into mitochondria by specific mitochondrial carriers [185]. The involvement of the mitochondrial transporters Mrs3p and Mrs4p in mitochondrial iron uptake was firstly investigated and characterized in *S. cerevisiae* [187,188]. Indeed, a clear correlation between Mrs3p and Mrs4p expression levels, mitochondrial iron levels and the efficiency of heme and Fe/S cluster synthesis was found in yeast [189,190]. This correlation was also observed in vitro in isolated mitochondria, strongly suggesting that these carriers are directly involved in mitochondrial iron uptake [188]. In vitro transport assays performed using sub-mitochondrial vesicles prepared from yeast mitoplasts have shown that Mrs3/4p mediate the transport of Fe^2+^ and Cu^2+^ across the inner mitochondrial membrane [191]. However, the precise mechanism of iron transport by Mrs3/4p is still poorly known.

Human mitoferrin-1 (also named as MFRN1), encoded by the *SLC25A37* gene, and mitoferrin-2 (named also as MFRN2), encoded by the *SLC25A28* gene, share ~35% sequence identity with the yeast Mrs3/4p and functionally complement the *MRS3*/*4Δ* yeast mutant in iron-depleted conditions [192,193,194]. Mitoferrin-1 is highly expressed in differentiating erythroid cells, whereas mitoferrin-2 is ubiquitously expressed at low levels in all non-erythroid tissues [193,194,195]. In human cells silenced for both mitoferrin-1 and -2, mitochondrial iron transport was found to be reduced more than 90%, suggesting that mitoferrins contribute to most of the mitochondrial iron supply [196,197]. In human erythroid cells, the loss of mitoferrin-1 could induce heme synthesis defects due to impaired mitochondrial iron uptake [194,198,199], causing erythropoietic protoporphyria [200,201]. Moreover, the knockdown of mitoferrin-2 resulted in a decreased mitochondrial iron content in human epithelial cell lines [64], and its depletion in murine fibroblasts affected heme synthesis and mitochondrial Fe/S cluster assembly [202]. A similar effect was also observed for mitoferrin-1 depletion, indicating that both isoforms contribute to mitochondrial iron transport in non-erythroid cells [202]. On the other hand, mitoferrin-2 was found to be unable to restore heme synthesis in developing erythroid cells. This evidence demonstrates that although mitoferrin isoforms are homologous, they exhibit distinct functional roles [197].

In *D. melanogaster*, a single mitoferrin gene (*CG4963*, also named *dmfrn*) was found to encode the only putative homolog of vertebrate mitoferrin-2 [203,204]. Since mitoferrin-1 has a specific role in human erythropoiesis, the lack of erythropoiesis in the fruit fly could justify the presence of only one mitoferrin gene. The CG4963 protein shares high sequence identity and similarity with the yeast Mrs3/4p. In particular, the putative substrate-binding site and contact sites, which recognize and discriminate between different substrates of the same classes or between the different classes of substrates, are identical in the Drosophila and yeast proteins [203]. Moreover, *CG4963* was able to complement the growth defect of a *MRS3*/*4Δ* yeast double mutant under low iron conditions, and its overexpression in insect cell culture resulted in dynamic alterations of cellular iron homeostasis, similarly to what was found in yeast [185], thus confirming that the Drosophila and yeast proteins perform similar functions. Studies on *D. melanogaster* mitoferrin (*dmfrn*) mutants showed that the *Dmfrn* mitochondrial iron carrier is essential for male fertility, playing a role in fruit fly spermatogenesis and development [203]. Indeed, Drosophila male hypomorphic mutants containing a transposable element insertion at specific locations in the 5′ untranslated region (UTR) of *dmfrn* presented severe defects in spermatogenesis leading to sterility. Moreover, male sterility was rescued in transgenic fly lines containing the whole genetic region of *dmfrn*, strongly suggesting that mitochondrial iron transport is crucial for spermatogenesis [203]. In particular, *dmfrn* mutant fruit flies exhibited abnormal testicular development characterized by unelongated or improperly elongated spermatids or by normally elongated spermatids but never-mature sperm. Mitochondria are known to be essential organelles of developing spermatids in Drosophila, strongly satisfying metabolic demand and supporting structural reorganization during spermatogenesis. In this context, it was hypothesized that a deficiency of mitochondrial functions could affect spermatid development [203]. Indeed, insufficient *dmfrn* expression could cause defects in the respiratory chain complexes containing heme or ISC as prosthetic groups, which are essential not only for energy production, since they are also directly involved in the spermatid individualization process [205]. The special role of mitochondrial iron metabolism in spermatogenesis and, consequently, the direct involvement of *dmfrn* were further supported by the high expression level found in the testes of the *dmfrn* protein and of other iron metabolism proteins, such as frataxin and mitochondrial ferritin. Importantly, this evidence was also found in mammals, suggesting that the intimate connection between mitochondrial iron metabolism and sperm function is highly conserved from insects to mammals [205,206]. Accordingly, a recent study reported that the loss of murine mitoferrin-2 function caused a significant decrease in male fertility in mice due to a dramatic reduction in sperm counts and motility, confirming a primary role of mitochondrial iron transport in mammalian spermatogenesis [207]. Similarly to what was found in *MRS3*/*4Δ* yeast double mutants, which presented a growth defect exclusively on low iron medium, the male fertility of hypomorphic *dmfrn* mutants was found to be related to the iron content of the food they were raised on. The supplementation of food with iron increased fertility but only in the weaker mutants characterized by low expression levels of *dmfrn* and by a less severe sterility phenotype. On the contrary, male fruit flies with the strong mutant phenotype were completely sterile on both low and high iron food, probably because *dmfrn* expression is too low to allow spermatogenesis [206]. Moreover, recently, green tea polyphenols were reported to play a role in maintaining iron homeostasis in fruit flies through the regulation of proteins involved in iron metabolism, such as *dmfrn* protein. Finally, it was found that *dmfrn* deletion or knocking down mutants compromised the development of larvae to adulthood only in the presence of low iron concentrations, whereas no lethality was observed on normal or high iron food, indicating that the lethality phenotype was enhanced in iron-deprivation spermatogenesis [208]. Moreover, this evidence, together with the established fundamental role of mitochondrial iron metabolism in viability, may suggest that the *dmfrn* protein may not represent the only mitochondrial iron transporter in Drosophila, similarly to what has been proposed for the yeast Mrs3/4p. In fact, in *S. cerevisiae*, the mitochondrial pyrimidine transporter Rim2p/Mrs12p [209] was shown to also have mitochondrial iron transport activity, being able to rescue the Mrs3/4p double deletion phenotype [210]. Likewise, in Drosophila, a protein encoded by the *CG18317* gene, known as drim2, was identified as a putative homolog of Rim2p/Mrs12p [211]. Preliminary studies have revealed that the CG18317 protein could have a similar function to the yeast Rim2p/Mrs12p and could be another low affinity iron carrier in Drosophila.

### 2.9. The Pyrimidine (Deoxy)Nucleotide Carrier: CG18317 vs. SLC25A33 and SLC25A36

In humans, the *SLC25A33* and *SLC25A36* genes encode two mitochondrial proteins named SLC25A33 and SLC25A36, respectively, sharing 59.8% identity [212]. Recombinant SLC25A33 and SLC25A36 overexpressed in bacterial cells and functionally reconstituted into liposomes transported pyrimidine deoxynucleotides.

In detail, SLC25A33 is an obligatory counter-exchanger, efficiently transporting uracil and thymine nucleoside di- and triphosphates, and, to a lesser extent, cytosine and guanine deoxynucleoside di- and triphosphates, (d)UTP and ITP, but it is unable to exchange nucleoside monophosphates [212]. SLC25A36 preferentially exchanges cytosine and inosine and, to a lesser extent, guanine and uracil deoxynucleoside mono-, di- and triphosphates. Notably, SLC25A36 is also able to catalyze uniport reactions [212]. Both carriers are severely inhibited by tannic acid, BAT, PLP, some mercurial reagents and bromocresol purple [212]. Since pyrimidine (d)NTPs are used during DNA and RNA synthesis, whereas pyrimidine (d)NMPs originate from their breakdown, SLC25A33 and SLC25A36 could play a key role in essential mitochondrial processes, including the synthesis and breakdown of mitochondrial DNA (mtDNA), as well as in its replication and repair [212]. Considering that SLC25A33 does not transport nucleoside monophosphates, its biological function could be the import of cytosolic pyrimidine (d)NTPs in exchange for intramitochondrial pyrimidine (d)NDPs synthesized in mitochondria by the conversion of pyrimidine (d)NMPs into their corresponding (d)NDPs [213]. SLC25A33 was previously named pyrimidine nucleotide carrier 1 (PNC1); in intact cultured human cells, PNC1 could mediate the mitochondrial trafficking of thymidine nucleotides, since its overexpression increased (d)TTP mitochondrial content, whereas its downregulation decreased the (d)TTP export rate [214]. PNC1 was also shown to promote cell growth and proliferation by affecting mitochondrial activity. In more detail, its overexpression increased cell size and impaired mitochondrial biogenesis, whereas its suppression reduced mitochondrial UTP levels and the oxidative phosphorylation rate, increased cellular ROS levels and delayed cell cycle progression and proliferation [215]. Additionally, in human cell lines, PNC1 was shown to maintain mtDNA and to affect its replication and transcription [215]. Notably, PNC1 was found to be overexpressed in cells growing very fast, such as transformed fibroblasts and various cancer cell lines, supporting the hypothesis that its activity could be required to promote tumor initiation and progression [215]. The biological role of SLC25A36 could be the import of pyrimidine (d)NTPs into the mitochondrial matrix in exchange for intramitochondrial pyrimidine (d)NMPs or, to a lesser extent, pyrimidine (d)NDPs, especially in quiescent and cycling cells, in which the contribution of the salvage pathway synthesizing nucleotides from the corresponding nucleosides in mitochondria is not enough [216]. Moreover, the supplementary contribution due to the uniport reaction catalyzed by SLC25A36 could be necessary when mitochondrial (d)NTPs need to be elevated, especially during mitochondrial biogenesis [212]. Interestingly, *SLC25A36* appeared to be overexpressed in some cases of cervical cancers [64]. *SLC25A36* was also highly expressed in naive mouse embryonic stem cells (mESCs), in which its suppression was proved to impair glutathione metabolism, to deplete mtDNA content and to affect mitochondrial morphology, leading to mitochondrial dysfunction and promoting mESC differentiation [217]. In *Saccharomyces cerevisiae*, the *Rim2* gene encodes a mitochondrial pyrimidine nucleotide antiporter known as Rim2p, able to exchange pyrimidine deoxynucleoside tri- and di-phosphates and, to a lesser extent, pyrimidine deoxynucleoside monophosphates; its deletion was found to cause mtDNA depletion, as well as yeast cells’ inability to grow on non-fermentable carbon sources [209]. In yeast cells lacking *Rim2,* the expression of *SLC25A33* or *SLC25A36* restored the yeast phenotypes, proving that these human carriers have transport functions comparable to those of Rim2p [212].

In the *D. melanogaster* genome, only one gene has been identified as having significant similarity to the yeast *Rim2* and the human *SLC25A33* and *SLC25A36* genes, i.e., *CG183173* or *drim2*. It originates three different transcripts corresponding to three putative protein isoforms sharing a structural organization typical of all the SLC25 family members. All the isoforms exhibit comparable sequence similarity (about 40%) to the yeast Rim2p sequence, as well as being 52–54% identical to the human SLC25A33/PNC1, as revealed by sequence alignment analysis [211]. The mitochondrial subcellular localization of the dRIM2 protein was confirmed by immunofluorescence assays performed on *D. melanogaster* S2R^+^ cells transiently transfected with a proper vector containing dRIM2 cDNA. Endogenous *drim2* expression was down-regulated in S2R^+^ cells, then the mitochondrial and cytosolic (d)NTP pool sizes were measured in both silenced and control cells. The silencing of *drim2* did not influence (d)NTP cytosolic contents, but, interestingly, silenced S2R^+^ cells exhibited significantly reduced mitochondrial concentrations of both purine and pyrimidine (d)NTPs, especially as regards (d)ATP and (d)TTP, suggesting that dRIM2 could be able to import all these DNA precursors into the mitochondrial matrix [211]. *D. melanogaster drim2* homozygous and heterozygous knock-out *(drim2^−/−^ and drim2^+/−^,* respectively) third instar larvae were generated in order to study their morphology, development and behavior [218]. Remarkably, *drim^−/−^* knock-outs were smaller than heterozygous insects; they did not reach adulthood (mostly dying during the larval stage) and exhibited severe locomotor defects, as well as their mitochondria displaying several abnormalities. In particular, *drim2^−/−^* mitochondria appeared disorganized and unable to line up along the z-lines, when examined by transmission electron microscopy. Additionally, these mitochondria had a rounder shape and a higher density with respect to those observed in *drim2^+/−^* insects. Furthermore, muscle body preparations of the *drim2^−/−^* third instar larvae showed decreased oxygen consumption rates, indicating an impairment of mitochondrial respiration. In this light, the observed higher mitochondrial density found in the *drim2^−/−^* knock-outs could be due to a compensatory response triggered by the mitochondrial failure revealed in these insects [211]. On the basis of such data, it is evident that dRIM2 is essential for preserving mitochondrial function in *D. melanogaster*, by providing the (d)NTPs required for mtDNA transcription and replication in the mitochondrial matrix [211]. In this light, dRIM2 could represent the *Drosophila* ortholog of the human *SLC25A33* and *SLC25A36*, as well as of the yeast Rim2p, on the basis of the high sequence identity and also considering that all these proteins have similar transport functions and their deficits can impair mitochondrial functions. It is feasible that dRIM2 could act as a general transporter of deoxynucleotides instead of being a typical pyrimidine carrier, because of the peculiar composition of the (d)NTP pool found in *D. melanogaster* cells. Indeed, they contain mostly (d)ATP, intermediate concentrations of (d)TTP and low amounts of (d)CTP and (d)GTP [211], whereas mammalian cells are rich in (d)TTP, have intermediate concentrations of (d)CTP and (d)ATP and have a low content of (d)GTP [219]. In support of this view, it is noteworthy that the yeast Rim2p preferentially transports thymine, uracil and cytosine nucleoside di- and triphosphates, and, to a lower extent, the corresponding guanine and adenine analogs [209], probably because the *S. cerevisiae* (d)NTP pool contains higher concentrations of (d)TTP and (d)CTP [220]. Notably, since a direct kinetic characterization of dRIM2 is still lacking, its real substrate specificity remains to be clarified (e.g., its ability to preferentially transport pyrimidine and/or purine (d)NTPs), as remain its functional similarities and differences with respect to the human SLC25A33 and SLC25A36.

### 2.10. The Shawn Protein: CG2616 and CG14209 vs. SLC25A39 and SLC25A40

The mammalian *SLC25A39* and *SLC25A40* genes encode two putative mitochondrial carriers designed SLC25A39 and SLC25A40, whose substrate specificity has not been determined by transport assays in reconstituted systems to date, so their biological roles are poorly understood.

In humans, these genes are expressed to a large extent in the brain [221] and reside independently in risk loci for epilepsy [222,223]. *SLC25A40* was associated with chronic fatigue in primary Sjogren’s syndrome [224], and it could play a role in hypertriglyceridemia [225]. *SLC25A39* was firstly identified in *S. cerevisiae* in an attempt to find activators of manganese-containing superoxide dismutase (SOD_2_), an enzyme endowed with high antioxidant activity needing manganese as a cofactor [226].

Briefly, mutations found in the yeast gene *YGR257c* (corresponding to *slc25a39*) specifically reduced the mitochondrial SOD_2_ activity, without affecting that of a cytosolic manganese-containing SOD [227]; hence, this yeast gene was called manganese trafficking factor for mitochondrial SOD_2_ (*MTM1*) [226]. Interestingly, *mtm1*Δ mutants did not show a decreased manganese content in mitochondria, but they contained very high iron levels able to compete with manganese to be incorporated into SOD_2_, thus affecting its activity. On this basis, mtm1p was believed to be a chaperone facilitating manganese insertion into the mitochondrial SOD_2_, thus specifically activating it [226]. In this regard, other studies performed on *S. cerevisiae* highlighted that SOD_2_ was able to preferentially incorporate manganese over iron according to their respective intramitochondrial levels [228]. Moreover, SOD_2_ activity was found to be impaired when mutations affected the activity of the same proteins related to mitochondrial iron homeostasis, including mtm1p [229]. A link between iron dyshomeostasis and mtm1p was also revealed in the heme biosynthesis pathway in murine erythroleukemia cells, in which a significant decrease in iron incorporation into protoporphyrin IX was found upon *Slc25a39* silencing, suggesting that this gene could play a role in regulating mitochondrial iron homeostasis, even if it was not directly involved in iron transport [230].

Further insights into *SLC25A39* and *SLC25A40*’s biological functions came from a genetic screen performed in *D. melanogaster*, aiming to identify genes involved in neuronal transmission [231]. *Shawn* was identified as a *D. melanogaster* homolog of the human *SLC25A39* and *SLC25A40* and the yeast *mtm1*, having 43%, 46% and 33% sequence identity, respectively. *Shawn* encodes a well-conserved SLC25 family member, which localizes to mitochondria and is ubiquitously expressed in each fruit fly development stage, even if at low levels [231].

In *shawn*, biallelic missense mutations were found in conserved sequence motifs shared by all the SLC25 family carriers, which are believed to be critical for transport function [11].

In the fruit fly, such mutations were proved to be responsible for the gradual neurodegeneration of photoreceptor terminals, as well as the degeneration of postsynaptic muscle cells at the larval neuromuscular junctions (NMJs), which implied mitochondrial dysfunction and cell death [231]. In more detail, in synaptic *shawn* mutant neurons, increased cytoplasmic levels of manganese were found, suggesting a defective cytoplasmic clearance of this metal, likely able to interfere with redox homeostasis. Furthermore, synaptic bouton mitochondria accumulated ROS and were dysfunctional; they had altered morphology and potential, along with unbalanced metal homeostasis. In this regard, significant increases in calcium and free-iron intramitochondrial concentrations were found. Notably, the increased pool of free iron in the mitochondrial matrix was able to affect the activity of some Fe/S enzymes, including aconitase [231]. Interestingly, defects in metal homeostasis were already known to increase oxidative stress, leading to neurotoxicity with a potentiation of glutamatergic transmission [232]. In the larval NMJs, a loss of *shawn* deeply altered the ultrastructure of muscle cells, whereas that of presynaptic terminals was unaffected. Such muscle cells were subjected to an excessive glutamatergic transmission from motor neurons and contained swollen mitochondria showing various autophagic profiles and large electron-dense inclusions, which were probable sites of calcium overload acting as pro-apoptotic signals. In this regard, it is likely that muscle cells could be more affected than neurons by increased oxidative stress levels, as found in rats [233]. Remarkably, a similar phenotype also implying iron toxicity was already observed in mutations affecting other *Drosophila* genes including pink1 or parkin, linked to Parkinson’s disease [234,235]. In *D. melanogaster*, these findings highlight that *shawn* could play a key role in modulating neurotransmitter release, preserving neuronal survival, as well as in preventing mitochondrial dysfunction, by affecting metal homeostasis and ROS generation. The human SLC25A39 and SLC25A40 mitochondrial carriers could be involved in similar pathways in the brain, considering that they reside in risk loci for epilepsy and that the potentiation of glutamatergic transmission might predispose to neurological disorders. The complete kinetic characterization of the human SLC25A39 and SLC25A40 proteins, as well as that of Shawn, would be necessary to clarify their transport properties, in order to fully understand their biological functions, as well as their possible involvement in the genesis of neurological diseases.

### 2.11. Dephosphocoenzyme A Carrier: CG4241 vs. SLC25A16 and SLC25A42

Coenzyme A (CoA) is a ubiquitous and essential cofactor involved in a lot of central metabolic reactions, including fatty acid β-oxidation, carbohydrate and amino acid oxidation, and the Krebs cycle. The compartmentation of CoA in all eukaryotes seems to be closely regulated, with the cytosol and organelles maintaining separate CoA pools, whose levels can modulate fluxes through CoA-dependent reactions [236,237].

The transport of CoA was firstly hypothesized in rats [238]. Successively, in humans, the transport of CoA across the inner mitochondrial membrane was ascribed to two related genes, i.e., *SLC25A16* and *SLC25A42*, but only the latter was functionally characterized. The transport properties and kinetic characteristics of the recombinant SLC25A42 protein have demonstrated that it mediates the transport of CoA into mitochondria in exchange for intramitochondrial adenosine 3’,5’-diphosphate (PAP) and (deoxy)adenine nucleotides [239]. Furthermore, dephospho-CoA (dPCoA) is also efficiently transported by SLC25A42, but it is not yet clear whether it is imported into the matrix, in which it could be converted into CoA, or if it could act as a counter substrate for CoA uptake [239]. In the mitochondrial matrix, CoA is essential for several processes, including the Krebs cycle, the synthesis and β-oxidation of fatty acids, amino acid catabolism and the urea cycle [239]. Moreover, mitochondrial CoA, together with the glycine imported into the mitochondria by its specific carrier SLC25A38, is required for heme biosynthesis, being involved in the first and rate-limiting step of this pathway [240].

On the contrary, SLC25A16, known as Graves disease carrier [2], was previously thought to be the human ortholog of the mitochondrial CoA carrierLeu5p in *S. cerevisiae*. Yeast leu5p-deficient (leu5Δ) cells, failing to accumulate mitochondrial CoA, were rescued by the expression of SLC25A16 [171]. However, this evidence is only indirect, because both the yeast and human protein were not directly proven to function as CoA transporters.

Moreover, some authors do not consider SLC25A16 and SLC25A42 to be isoforms of the same carrier, because of their too-low sequence identity and similarity (34% and 46%, respectively). Indeed, it is known that isoforms of mitochondrial transporters share at least 55% identity [1]. SLC25A42 and SLC25A16 are likely to show differences in their transported substrates.

In the *D. melanogaster* genome, only one gene is present, *CG4241*, phylogenetically close to *SLC25A42*. *CG4241* encodes two alternatively spliced isoforms; i.e., the functionally characterized dPCoAC-A catalyzes the transport of dPCoA and, to a lesser extent, ADP and dADP (but not CoA), whereas the recombinant dPCoAC-B protein has displayed no transport activity under any of the experimental conditions tested [25]. However, studies on the phenotype complementation of the yeast leu5Δ strain revealed that both dPCoACs restored the growth defect of the *S. cerevisiae* leu5Δ strain on non-fermentable carbon sources, as well as mitochondrial CoA content, demonstrating that Drosophila carriers are able to translocate dPCoA instead of CoA [25].

SLC25A42 and dPCoAC-A appear to have distinct functions, showing important differences in their substrate specificity. In particular, the human carrier transports both CoA and dPCoA at low but significant rates, whereas the fruit fly ortholog has the highest substrate affinity for dPCoA, but it does not transport CoA. Moreover, only SLC25A42 efficiently transports PAP [25,239].

These data suggest a possible physiological role of dPCoAC-A in the regulation of CoA metabolism and compartmentalization in the fruit fly. In fact, when CoA levels in the mitochondrial matrix increase compared to those in the cytosol, CoA is converted into dPCoA, which is transported by dPCoAC-A out of the mitochondria in exchange for ADP. In the cytosol, it is phosphorylated to CoA, thus keeping constant the two CoA pools.

Furthermore, relevant differences in the amino acid composition of the two isoforms were also observed in the residues of the matrix gate (m-gate) area; this could explain their different substrate specificity. In fact, these residues can interact with the transported substrate and are directly involved in the conformational changes required during substrate translocation [2].

A structural analysis was conducted on comparative models of dPCoAC-A and SLC25A42, in the presence of CoA and dPCoA as ligands. It revealed that both isoforms can bind CoA and dPCoA in a very similar way, with only a relevant difference. In this regard, in dPCoAC-A, one ionic interaction is present between the arginine residue at position 103 and the phosphate group of the CoA ribose ring, but this bond is absent in SLC25A42 since R103 is replaced by a serine. It seemed possible that this residue could be involved in specific conformational changes preventing CoA translocation through dPCoAC-A [25]. This hypothesis was confirmed in a dPCoAC-A mutant, in which this arginine was replaced by a serine residue. The resulting mutant proved to be able to translocate CoA, unlike the wild-type protein, demonstrating that the residue in position 103 is directly involved in substrate uptake in the mitochondrial matrix [25].

### 2.12. CG8931 and CG5755 vs. SLC25A46

In humans, some mitochondrial carrier family members can be considered orphan transporters, since their transported substrates are still unknown. Among these, SLC25A46 seems to have no transport function and, differently from the other mitochondrial carriers, it localizes to the outer mitochondrial membrane. Recently, SLC25A46 was supposed to be implicated in both mitochondrial dynamics and crista morphology, being identified as the closest human homolog to Ugo1, a yeast protein involved in mitochondrial fission [241,242]. Moreover, the knockdown of this protein in human cultured cells caused mitochondrial hyperfusion due to decreased mitochondrial fission [241]. Altered mitochondrial and endoplasmic reticulum morphology, destabilization of the MICOS (mitochondrial contact site and cristae organizing system) complex, impaired cellular respiration and premature cellular senescence were also observed [242].

In the last years, SLC25A46 has attracted worldwide scientific interest because of its involvement in the onset of several mitochondrial neurological diseases, including inherited optic atrophy, Charcot-Marie-Tooth type 2, Leigh syndrome, progressive myoclonic ataxia and lethal congenital pontocerebellar hypoplasia. Indeed, several mutations found in the *SLC25A46* gene have been identified and related to these neurological disorders [243,244,245,246,247,248].

In *D. melanogaster*, two genes have been identified as human *SLC25A46* homologs, *CG8931* (named dSLC25A46a) and *CG5755* (named dSLC25A46b); dSLC25A46a, if compared to dSLC25A46b, displays higher homology with the human *SLC25A46*. Their physiological role has not yet been clarified, but they are predicted to be involved in mitochondrial membrane fission and in mitochondrial dynamics, like the human homolog [249]. In this regard, recently developed dSLC25A46a or dSLC25A46b knockdown Drosophila models displayed most of the phenotypes already observed in mitochondrial diseases caused by human *SLC25A46* mutations. In particular, locomotor impairment and defects in neuromuscular junctions compromising synaptic function were observed in mutant fruit flies at different developmental stages. The knockdown of dSLC25A46a or dSLC25A46b also led to severe structural and functional mitochondrial defects, at least in part associated with ROS accumulation and ATP level reductions [249,250].

The phenotypes induced by the knockdown of each dSLC25A46 gene seem to be very similar, suggesting that they might play non-superimposable roles in *D. melanogaster*, not complementing the defects induced by the knockdown of each other. Moreover, the subcellular localization of dSLC25A46b supports this hypothesis, since it localizes not only to mitochondria but also to the plasma membrane, where it could perform a different range of functions [250].

## 3. Conclusions

Mitochondrial carriers are membrane proteins transporting across mitochondria many different metabolites, which are involved in several biochemical pathways, including energetic metabolism, hormonal metabolic homeostasis, cell survival, proliferation and protection from oxidative stress, among many others. This review summarizes the main features and biological activities of the mitochondrial carriers characterized to date in *D. melanogaster*, highlighting their similarities and differences with their human counterparts. This may result in a better understanding of their physiological roles, also in light of some important Drosophila pathways found to be modulated by these transporters, including fruit fly phenotype, development, fertility, chromosomal integrity, lifespan extension and survival.

In the current state of knowledge, only some Drosophila mitochondrial carriers have been heterologously overexpressed, purified and kinetically characterized, often presenting transport features very similar to those of their corresponding human proteins. The functional role of other mitochondrial fruit fly carriers remains to be ascertained. For this aim, in the future, a full kinetic characterization in reconstituted proteoliposomes would be useful to shed light on their biological role, probably paving the way for the discovery of new hypothetical pathways also modulated by some of them in humans. Furthermore, the mutations found in some Drosophila genes result in models of genetic diseases such as Down syndrome, adPEO, Parkinson’s disease, epilepsy, inherited optic atrophy, Charcot-Marie-Tooth type 2, Leigh syndrome, progressive myoclonic ataxia and lethal congenital pontocerebellar hypoplasia. Interestingly, all of these diseases lead to impaired mitochondrial integrity and activity, and are somehow linked to the function and/or expression of important fruit fly mitochondrial carriers. In this light, Drosophila could represent an effective model system for investigating the molecular pathology of human diseases related to such carriers.

## Figures and Tables

**Figure 1 ijms-21-06052-f001:**
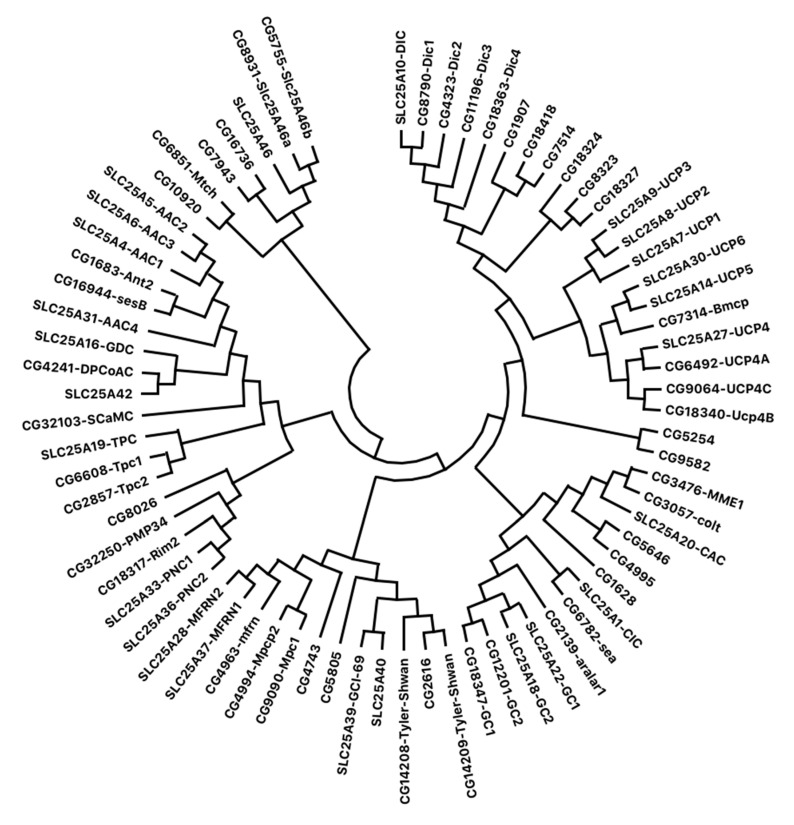
Phylogenic tree of amino acid sequences of mitochondrial carriers from *D. melanogaster* and *H. sapiens.* For comparative purposes, the 48 members of *Drosophila melanogaster* (Appendix A) and the 25 human members reported in Appendix A were employed. Sequences were retrieved from the GenBank, EMBL and Flybase databases. The evolutionary history was inferred by the use of the Maximum Likelihood method and JTT matrix-based model [15]. The tree with the highest log likelihood (−49,468.35) is displayed. Initial tree(s) for the heuristic search were automatically obtained by applying Neighbor-Join and BioNJ algorithms to a matrix of pairwise distances estimated using the JTT model and then selecting the topology with a superior log likelihood value. The tree is drawn to scale, with branch lengths representing the number of substitutions per site. This analysis included 73 amino acid sequences. The final dataset comprised a total of 910 positions. Evolutionary analyses were performed in MEGA X [2,3].

**Table 1 ijms-21-06052-t001:** Members of the human and *Drosophila melanogaster* mitochondrial carrier family.

Human Gene Name	Protein Name	Predominant Substrates	Splice Variants/Polypeptides	Drosophila Gene Name	Protein Name	Predominant Substrates	Splice Variants/Polypeptides	Sequence Homology
*SLC25A1*	CIC	Citrate, isocitrate, malate, PEP	2 mRNA/2 polypeptides	*CG6782*	sea (scheggia), DmCIC	Citrate, isocitrate, malate, PEP	4 mRNA/1 polypeptide	61%
*SLC25A4*	AAC1, ANT1	ADP, ATP		*CG1683*	ANT2	ADP, ATP	3 mRNA/1 polypeptide	71%
				*CG16944*	SesB	ADP, ATP	5 mRNA/2 polypeptides	78%
*SLC25A5*	AAC2, ANT2	ADP, ATP		*CG1683*	ANT2	ADP, ATP	3 mRNA/1 polypeptide	70%
				*CG16944*	SesB	ADP, ATP	5 mRNA/2 polypeptides	80%
*SLC25A6*	AAC3, ANT3	ADP, ATP		*CG1683*	ANT2	ADP, ATP	3 mRNA/1 polypeptide	70%
				*CG16944*	SesB	ADP, ATP	5 mRNA/2 polypeptides	79%
*SLC25A7*	UCP1	H^+^		*CG6492*	DmUCP4A	H^+^	3 mRNA/2 polypeptides	28%
				*CG18340*	DmUCP4B	H^+^	3 mRNA/3 polypeptides	27%
				*CG9064*	DmUCP4C	H^+^		27%
				*CG7314*	Bmcp, DmUCP5	H^+^	2 mRNA/1 polypeptide	30%
*SLC25A8*	UCP2	malate, oxaloacetate, sulfate, phosphate, aspartate		*CG6492*	DmUCP4A	H^+^	3 mRNA/2 polypeptides	31%
				*CG18340*	DmUCP4B	H^+^	3 mRNA/3 polypeptides	28%
				*CG9064*	DmUCP4C	H^+^		26%
				*CG7314*	Bmcp, DmUCP5	H^+^	2 mRNA/1 polypeptide	32%
*SLC25A9*	UCP3	H^+^	2 mRNA/2 polypeptides	*CG6492*	DmUCP4A	H^+^	3 mRNA/2 polypeptides	29%
				*CG18340*	DmUCP4B	H^+^	3 mRNA/3 polypeptides	30%
				*CG9064*	DmUCP4C	H^+^		28%
				*CG7314*	Bmcp, DmUCP5	H^+^	2 mRNA/1 polypeptide	31%
*SLC25A10*	DIC	malate, phosphate, succinate, sulphate, thiosulphate		*CG8790*	Dic1	malate, phosphate, succinate, sulphate, thiosulphate	3 mRNA/1 polypeptide	57%
				*CG4323*	Dic2	N/D		47%
				*CG11196*	Dic3	phosphate, sulphate, and thiosulphate	2 mRNA/2 polypeptides	45%
				*CG18363*	Dic4	N/D	3 mRNA/2 polypeptides	35%
*SLC25A14*	UCP5, BMCP1	sulfate, thiosulfate, sulfite, l-malate, malonate, maleate, phosphate, oxalate, l-citramalate, d-citramalate	3 mRNA/3 polypeptides	*CG6492*	DmUCP4A	H^+^	3 mRNA/2 polypeptides	30%
				*CG18340*	DmUCP4B	H^+^	3 mRNA/3 polypeptides	29%
				*CG9064*	DmUCP4C	H^+^		27%
				*CG7314*	Bmcp, DmUCP5	H^+^	2 mRNA/1 polypeptide	49%
*SLC25A16*	GDC (Graves’ disease carrier)	N/D		*CG4241*	dPCoAC	dPCoA, ADP, dADP	6 mRNA/3 polypeptides	30%
*SLC25A18*	GC2	Glutamate		*CG18347*	DmGC1	Glutamate	2 mRNA/1 polypeptide	50%
				*CG12201*	DmGC2	Glutamate	2 mRNA/2 polypeptides	45%
*SLC25A19*	DNC, TPC	Thiamine pyrophosphate, thiamine monophosphate, (d)NTPs	3 mRNA/1 polypeptide	*CG6608*	Tpc1	Thiamine pyrophosphate, (d)NTPs	2 mRNA/1 polypeptide	34%
				*CG2857*	Tpc2	Thiamine pyrophosphate, (d)NTPs		31%
*SLC25A20*	CAC	Carnitine, acylcarnitine		*CG3057*	Colt	Carnitine, acylcarnitine	2 mRNA/1 polypeptide	50%
				*CG3476*	MME1	Mg^2+^		44%
*SLC25A22*	GC1	Glutamate		*CG18347*	DmGC1	Glutamate	2 mRNA/1 polypeptide	54%
				*CG12201*	DmGC2	Glutamate	2 mRNA/2 polypeptides	48%
*SLC25A27*	UCP4	H^+^		*CG6492*	DmUCP4A	H^+^	3 mRNA/2 polypeptides	51%
				*CG18340*	DmUCP4B	H^+^	3 mRNA/3 polypeptides	42%
				*CG9064*	DmUCP4C	H^+^		34%
				*CG7314*	Bmcp, DmUCP5	H^+^	2 mRNA/1 polypeptide	32%
*SLC25A28*	Mitoferrin 2, Mfrn2	Fe^2+^		*CG4963*	Mitoferrin, dmfrn	Fe^2+^		41%
*SLC25A30*	UCP6, KMCP1	sulfate, thiosulfate, sulfite, l-malate, malonate, maleate, phosphate, oxalate, l-citramalate, d-citramalate		*CG6492*	DmUCP4A	H^+^	3 mRNA/2 polypeptides	31%
				*CG18340*	DmUCP4B	H^+^	3 mRNA/3 polypeptides	30%
				*CG9064*	DmUCP4C	H^+^		26%
				*CG7314*	Bmcp, DmUCP5	H^+^	2 mRNA/1 polypeptide	57%
*SLC25A31*	AAC4, ANT4	ADP/ATP		*CG1683*	ANT2	ADP, ATP	3 mRNA/1 polypeptide	65%
				*CG16944*	SesB	ADP, ATP	5 mRNA/2 polypeptides	68%
*SLC25A33*	PNC1	UTP		*CG18317*	Rim2	dNTPs	3 mRNA/3 polypeptides	43%
*SLC25A36*	PNC2	Pyrimidine nucleotides	2 mRNA/2 polypeptides	*CG18317*	Rim2	dNTPs	3 mRNA/3 polypeptides	47%
*SLC25A37*	Mitoferrin 1, Mfrn1	Fe^2+^		*CG4963*	Mitoferrin, dmfrn	Fe^2+^		53%
*SLC25A42*		CoA, adenosine 3’,5’-diphosphate		*CG4241*	dPCoAC	dPCoA, ADP, dADP	6 mRNA/3 polypeptides	47%

Note: N/D, not defined.

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
