# Peer review of "Drosophila melanogaster Mitochondrial Carriers: Similarities and Differences with the Human Carriers"

_ijms, 2020, doi:10.3390/ijms21176052_

Round 1
Reviewer 1 Report
In this manuscript Rosita Curcio et al. reviewed the current knowledge of mitochondrial carriers in Drosophila melanogaster in comparison to human and mammals. Overall, the manuscript are well organized, appropriately described, and the manuscript is well written.
Minor
1) Page 2, Line 84: In table I should be replaced with table 1.
2) Page 3, Line 149: In this section, author mainly used AAC. I believe ANT2 should explain like ANT2 (ACC2).
3) Page 11, Line 138: Drosophila melanogaster -> D. melanogaster
Author Response
Please see the attachment.
Response to Reviewer 1 Comments
point 1
Page 2, Line 84: In table I should be replaced with table 1
response: it has been corrected
point 2
Page 3, Line 149: In this section, author mainly used AAC. I believe ANT2 should explain like ANT2 (ACC2).
response: for more clarity ANT2 has been substituted with AAC2
point 3
Page 11, Line 138: Drosophila melanogaster -> D. melanogaster
response: it has been done
Reviewer 2 Report
The review manuscript "Drosophila melanogaster mitochondrial carriers: similarities and differences to the human carriers" provides a thorough analysis of the various mitochondrial carriers and a comprehensive description of their kinetics, function and role in health and disease establishing a nice parallel to their corresponding Drosophila homologs/orthologs.
This is a very interesting review, well written and easy to read. I have revised this paper from the perspective of Drosophila genetics and mitochondrial function, and I have a few minor concerns:
- Introduction: lines 100-125. Several statements in this section are missing appropriate references to support them. Please insert appropriate citations. Additionally, transition between line 112 through 115 looks chopped. Authors wrapped the idea in line 112 to touch on the subject again in line 113 in a different paragraph. I think these statements should belong to the same paragraph.
- subtitles 2.1 and 2,2 under subtitle: "Drosophila melanogaster versus human mitochondrial carriers". The opening of these two subsections focuses on the mammalian carriers and not specifically on the human ones. Authors should reconsider rearranging the paragraphs (particularly section 2.1) to focus the review mainly on the human mitochondrial carriers at the beginning, and take the time to expand further on the comparisons fly-human. Additional information on other mammalian carriers (ie. fly-rodent comparison before discussing the human carrier in subsection 2.1 line 102-103) are relevant but should not be the focus of the discussion in this review. Author should consider this recommendation and reorganize their statements/paragraphs accordingly. Same is evident in subsection 2.2 (lines 147-148).
- Subsections 2.2 and 2.3 would benefit from an overall summary statement at the end of the subsection.
Author Response
"Please see the attachment"
Red text in the manuscript file indicates changes made in response to the suggestions of Reviewer.
Point 1
Introduction: lines 100-125. Several statements in this section are missing appropriate references to support them. Please insert appropriate citations. Additionally, transition between line 112 through 115 looks chopped. Authors wrapped the idea in line 112 to touch on the subject again in line 113 in a different paragraph. I think these statements should belong to the same paragraph.
response:
According to the Reviewer suggestion, we have added appropriate references. Furthermore, as suggested, we have reorganized we rearranged the paragraph between the lines112-115 (introduction).
Point 2 subtitles 2.1 and 2,2 under subtitle: "Drosophila melanogaster versus human mitochondrial carriers". The opening of these two subsections focuses on the mammalian carriers and not specifically on the human ones. Authors should reconsider rearranging the paragraphs (particularly section 2.1) to focus the review mainly on the human mitochondrial carriers at the beginning, and take the time to expand further on the comparisons fly-human. Additional information on other mammalian carriers (ie. fly-rodent comparison before discussing the human carrier in subsection 2.1 line 102-103) are relevant but should not be the focus of the discussion in this review. Author should consider this recommendation and reorganize their statements/paragraphs accordingly. Same is evident in subsection 2.2 (lines 147-148).
response:
According to the Reviewer suggestion, we have reorganized the subsections 2.1 and 2.2.
Point 3 Subsections 2.2 and 2.3 would benefit from an overall summary statement at the end of the subsection.
response:
According to the Reviewer suggestion, we added a summary at the end of the subsections 2.2 and 2.3.